# TRAMBA: MAMBA WITH ADAPTIVE ATTENTION FOR TRAFFIC SPEED FORECASTING

## ABSTRACT

We introduce **Tramba**, a novel deep learning model for traffic speed forecasting in complex urban road networks. Unlike conventional methods that rely heavily on short-term trends or local spatial proximity (e.g., upstream and downstream links), Tramba captures dynamic, long-range dependencies across both time and space. It does so by integrating two key components: a Mamba-based temporal encoder that models long-term historical patterns of the target link, and an adaptive attention mechanism that learns temporally similar patterns from non-adjacent road links across the network. We evaluate Tramba on a real-world dataset from Gangnam-gu, Seoul, comprising 5-minute interval speed measurements across 366 road segments. Tramba is tested over forecasting horizons from 1 to 36 steps and compared with six strong baselines. It consistently outperforms all alternatives, achieving an average MAPE of 11.47%, MAE of 3.19 km/h, and MSE of 25.18 $(km/h)^2$ on TOPIS datasets for 12-step forecasting. These results highlight Tramba's ability to model long-range dependencies and detect non-local influences in complex urban networks, reducing prediction lag and improving robustness in dynamic traffic conditions. Code is available at https://github.com/tr-anon-users/tramba-code.

## 1 INTRODUCTION

Accurate traffic speed forecasting plays a critical role in intelligent transportation systems, supporting key functions such as congestion mitigation, adaptive signal control, and real-time route planning Ma et al. (2023). As with many problems, effective forecasting relies on the ability to capture long-range temporal dependencies and underlying traffic dynamics. Urban traffic patterns often evolve gradually over several hours, shaped by recurring commuting peaks, event-driven surges, or slow shifts in road capacity. Models that rely heavily on recent observations tend to lag behind actual transitions, an issue commonly referred to as prediction lag Min et al. (2023). This persistent challenge underscores the need for long-term memory capable of retaining historical traffic patterns Emami et al. (2019); Chen et al. (2022).

However, temporal modeling alone is insufficient. Many forecasting approaches improve performance by incorporating patterns from physically adjacent links (road segments), such as upstream or downstream links Kumar (2017); Xu et al. (2017). While this assumption may hold in highway networks with uniform flow conditions, it often fails in urban environments, where road links differ significantly in control schemes and geometric design Liu et al. (2023); Jia et al. (2016). In such settings, a link's traffic behavior may more closely align with that of a distant link sharing similar control logic, rather than with its immediate neighbors Yu et al. (2017); Dai et al. (2019); Wu et al. (2020). In these heterogeneous networks, physical proximity does not necessarily imply behavioral similarity Yu et al. (2017); Dai et al. (2019); Wu et al. (2020). This motivates the need for models that capture non-local dependencies by attending to distant yet temporally aligned links.

To address these challenges, we propose **Tramba**, a forecasting framework that captures both long-range temporal and non-local spatial dependencies in urban traffic. Tramba forecasts future speeds by retaining long-term patterns within each target link and selectively referencing time-shifted sequences from other links with similar historical behavior, regardless of spatial proximity. The temporal module uses a Mamba block for efficient sequence modeling over extended horizons, while the spatial module employs adaptive attention to identify relevant non-local influences based on tem-

poral similarity. This design allows Tramba to model complex spatiotemporal patterns and deliver accurate forecasts in large-scale urban networks. Our contributions can be summarized as follows:

- We propose **Tramba**, a novel time-series forecasting framework that integrates a selective state-space model with an adaptive attention mechanism for enhanced predictive accuracy and dynamic spatial reasoning in complex urban networks.

- Within **Tramba**, we propose two key modules: (1) a Mamba model that learns each link's temporal sequence for long-term forecasting, and (2) an adaptive attention module that compares all links to capture non-local spatial dependencies using temporal similarity.

- Extensive experiments on large-scale urban traffic datasets demonstrate that **Tramba** consistently outperforms strong baselines by better capturing the underlying spatiotemporal dynamics and reducing the reliance on short-term or lagged inputs.

## 2 RELATED WORK

### 2.1 URBAN TRAFFIC PREDICTION

Traditional time series models, such as auto-regressive integrated moving average and Kalman filtering, have been applied to short-term forecasting tasks, but their limited capacity to capture non-linear spatio-temporal dependencies has led to the rise of machine learning and deep learning approaches Emami et al. (2019); Kumar (2017); Xu et al. (2017). To improve the accuracy of urban traffic speed forecasting, various neural network-based models have been proposed Das et al. (2023). Lv et al. Lv et al. (2014), Yu et al. Yu et al. (2017), and Zhang et al. Dai et al. (2019) proposed deep neural network models, including deep belief networks, long short-term memory, and gated recurrent units, for capturing temporal dependencies. In addition, hybrid architectures that combine graph convolutional networks with attention mechanisms have demonstrated strong performance in modeling complex urban mobility patterns Wu et al. (2020). Cao et al. Cao et al. (2024) introduced multi-scale graph convolutional networks (GCNs) to capture diverse spatial dependencies; Hao et al. Li et al. (2024) proposed STFGCN, a spatio-temporal fusion GCN, for traffic forecasting; Bai et al. Bai et al. (2020) and Wu et al. Wu et al. (2020) utilized graph neural networks to model multivariate urban traffic dynamics. More recently, time-series exclusive methods were developed to capture time series patterns Xu et al. (2020); Lin et al. (2024); Oreshkin et al. (2019); Wang et al. (2024). Transformer-based architectures such as LCDFormer Cai et al. (2024), bidirectional spatial-temporal Transformer Chen et al. (2022), Pyraformer Liu et al. (2022) iTransformer Zou et al. (2024) and LSTTN Luo et al. (2024) have effectively captured both short- and long-term time series patterns. Transformer models leverage the self-attention mechanism, which allows them to dynamically weigh the relevance of each timestamp in relation to others Zeng et al. (2023); Nie et al. (2022). For example, Xu et al. Xu et al. (2020) incorporated spatial topologies into temporal modeling, while Informer Zhou et al. (2021) and Fedformer Zhou et al. (2022) enhanced robustness in long-horizon forecasting. These studies highlight the limitations of assuming upstream–downstream dependencies in urban contexts. Urban traffic is shaped by external controls and congestion patterns extending beyond adjacent areas, underscoring the need to model dynamic spatio-temporal relationships.

### 2.2 SELECTIVE STATE SPACE MODEL

The selective state space model (SSM), recently popularized through the Mamba architecture Gu & Dao (2023); Wang et al. (2025), has emerged as an efficient alternative to Transformer-based sequence models. Unlike Transformers, which relies on global attention mechanisms with quadratic complexity, Mamba adopts a different approach Ahamed & Cheng (2024); Gu et al. (2021). It introduces data-dependent parameterization and discretization within SSMs Lee et al. (2024). This enables long-range temporal modeling with near-linear computational complexity. Mamba was originally developed for natural language processing and computer vision tasks Gu & Dao (2023); Ma et al. (2024). Since then, it has been extended to time series forecasting across various domains Min et al. (2023). Its core mechanism combines convolutional processing, selective gating, and discretized dynamics Liang et al. (2024); Zeng et al. (2024). This allows the model to encode temporal patterns more efficiently than conventional recurrent or attention-based methods Xu et al. (2024). In the transportation domain, several studies have recently adopted Mamba-based

architectures to forecast traffic time-series patterns. For example, SOR-Mamba Lee et al. (2024) introduced a sequential order-robust version. This model improved robustness against input permutation and temporal noise. CMamba Zhang & Yan (2023) enhanced multivariate forecasting by modeling channel-wise correlations using Mamba blocks. Bi-Mamba+ Liang et al. (2024) proposed a bidirectional Mamba structure. This design allowed symmetric sequence modeling with improved spatial symmetry. DST-Mamba He et al. (2025) applied a temporal decomposition strategy. These studies highlight Mamba's strength in modeling temporal dependencies. However, in the context of traffic forecasting, time-series models still often suffer from prediction lag, where recent patterns are simply reproduced with a delay rather than capturing the true timing of transitions.

## 2.3 IMPLICATIONS

Recent traffic speed forecasting models primarily focus on temporal patterns or univariate sensor data Gu & Dao (2023); Wang et al. (2025), often neglecting spatial dependencies or simplifying them using static adjacency matrices derived from road topology Liang et al. (2024); He et al. (2025). Such fixed structures limit the model's ability to capture dynamic, context-aware spatial interactions Ali et al. (2024); Han et al. (2024). Moreover, standard global attention mechanisms do not account for temporal misalignment, making them ineffective at capturing the "time-shifted yet spatially distant" patterns common in real traffic networks Cai et al. (2024); Chen et al. (2022); Liu et al. (2022). These challenges underscore the need for parallel spatial and temporal modeling Emami et al. (2019); Kumar (2017); Xu et al. (2017). **Tramba** advances this direction by coupling a Mamba-based temporal backbone with shift-aware adaptive attention, enabling the model to learn dynamic, non-adjacent influences across the network.

## 3 METHODS

### 3.1 MODEL OVERVIEW

**Tramba** is a specialized deep learning model designed for time series tasks such as urban traffic speed forecasting. It comprises three main components: 1) a Mamba-based temporal module, 2) an adaptive attention mechanism for spatial learning, and 3) a sigmoid fusion gate. As shown in Figure 1, the model takes as input a sequence of traffic speeds from multiple road links and encodes them into latent representations using aggregation and adaptive embedding. These encoded features are then processed in parallel: the Mamba module captures long-range temporal dependencies within each target link through a SSM; the adaptive attention module computes relevance scores by comparing the current state of the target link with the historical patterns of all other links, based on a learnable similarity function. The outputs are co-integrated through an adaptive gating mechanism, which dynamically emphasizes temporal continuity or spatial similarity depending on the prediction context. The final output is a prediction of future traffic speeds for all links.

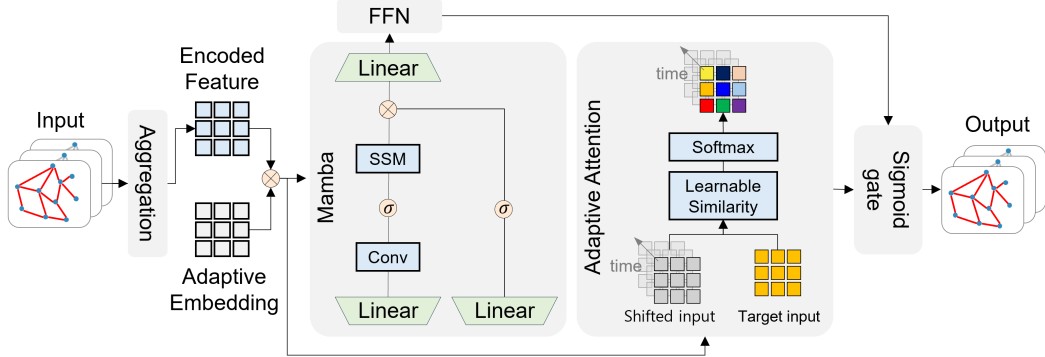

Figure 1: **Tramba Architecture.** The Tramba model processes input traffic sequences through two parallel modules: a Mamba block for modeling long-range temporal patterns, and an Adaptive Attention module that captures non-local spatial dependencies using time-shifted similarity. The outputs from both modules are fused via a sigmoid gate.

## 3.2 CORE COMPONENTS

**Input Encoding.** Each input tensor $\mathbf{X}_{\text{in}} \in \mathbb{R}^{B \times T \times L \times F}$, which contains historical speed sequences from multiple links, is projected into a latent space through a series of linear layers. Then a positional encoding is added to preserve temporal order and enhance temporal awareness. The final encoded tensor $\mathbf{X}$ maintains the original spatiotemporal structure, calculated as:

$$\mathbf{X} = \text{Linear}_{\text{adaptive}}(\text{Linear}_{\text{agg}}(\text{Linear}_{\text{embed}}(\mathbf{X}_{\text{in}}) + \text{PosEmb}(T))) \in \mathbb{R}^{B \times T \times L \times D}. \quad (1)$$

**Mamba Block.** To model long-range temporal dependencies, **Tramba** applies a selective SSM per link, enabling fine-grained temporal modeling without spatial interference. The SSM output is modulated by a learnable gate and refined via a residual feedforward layer, computed as:

$$\hat{\mathbf{Y}} = \text{SSM}(\mathbf{X}), \quad \hat{\mathbf{Y}} \in \mathbb{R}^{B \times T \times L \times D},$$
$$\mathbf{H}_{\text{mamba}} = \text{FFN}\Big(\hat{\mathbf{Y}} \odot \sigma(\text{Linear}(\mathbf{X}))\Big) + \hat{\mathbf{Y}}. \quad (2)$$

**Adaptive Attention.** **Tramba** uses an adaptive attention mechanism, introducing a learnable similarity function that uncovers temporal alignments across non-adjacent links. A module computes relevance scores across both spatial and temporal dimensions, which are then normalized into attention weights. This allows attention to temporally aligned yet distant links, computed as:

$$\mathbf{H}_{\text{attn},i} = \sum_{j=1}^{L} \sum_{s \in \mathcal{S}} \text{softmax}\Big(\text{Sim}_\theta(\mathbf{X}_i, \mathbf{X}_j^{(s)})\Big) \cdot \mathbf{X}_j^{(s)}. \quad (3)$$

The detailed computation steps for the Mamba block and the Adaptive Attention mechanism are summarized in Algorithm 1 and Algorithm 2, respectively.

---

**Algorithm 1** MambaBlock

**Require:** $\mathbf{X} \in \mathbb{R}^{B \times T \times D}$
1: Apply Conv1D + SiLU: $\mathbf{x}'$
2: Compute $\bar{\mathbf{A}}, \bar{\mathbf{B}}, \mathbf{C}$ via projection
3: Update state: $\mathbf{y}_t = \bar{\mathbf{A}}\mathbf{y}_{t-1} + \bar{\mathbf{B}}\mathbf{x}_t$
4: Project output: $\hat{\mathbf{y}}_t = \mathbf{C}\mathbf{y}_t$
5: Compute gate: $\mathbf{z} = \sigma(\text{Linear}(\mathbf{x}'))$
6: Modulate: $\mathbf{y}' = \hat{\mathbf{y}}_t \odot \mathbf{z}$
7: Apply FFN: $\mathbf{H}_{\text{mamba}} = \text{FFN}(\mathbf{y}') + \mathbf{y}'$
8: **return** $\mathbf{H}_{\text{mamba}}$

---

**Algorithm 2** AdaptiveAttention

**Require:** $\mathbf{X} \in \mathbb{R}^{B \times T \times L \times D}$
1: Generate shifted inputs: $\mathbf{X}^{(s)}$ for $s \in \mathcal{S}$
2: Compute similarity:
$\quad S_{i,j}^{(s)} = \text{LearnableSim}(\mathbf{X}_i, \mathbf{X}_j^{(s)})$
3: Normalize weights: $\alpha_{i,j}^{(s)} = \text{softmax}(S_{i,j}^{(s)})$
4: Apply attention: $\mathbf{H}_{\text{attn},i} = \sum_{j,s} \alpha_{i,j}^{(s)} \cdot \mathbf{X}_j^{(s)}$
5: **return** $\mathbf{H}_{\text{attn}}$

---

**Fusion and Output.**

**Fusion and Output.** At the final stage, temporal and spatial representations are integrated through a learnable gate $\boldsymbol{\omega}$ that adaptively emphasizes long-term memory or non-local similarity:

$$\mathbf{H}_{\text{fused}} = \boldsymbol{\omega} \odot \mathbf{H}_{\text{attn\_last}} + (1 - \boldsymbol{\omega}) \odot \mathbf{H}_{\text{mamba\_last}}. \quad (4)$$

A direct multi-horizon head then projects $\mathbf{H}_{\text{fused}}$ into predictions for all future steps:

$$\hat{\mathbf{Y}}_{1:\tau} = \text{Linear}_\tau(\mathbf{H}_{\text{fused}}). \quad (5)$$

Through this fusion-and-prediction design, **Tramba** unifies long-term temporal continuity and non-local spatial dynamics in a single forward pass, enhancing efficiency and accuracy.

**Tramba.** As shown in Figure 1, **Tramba** integrates the Mamba block and the Adaptive Attention module into a unified framework that captures both long-range temporal dependencies and non-local spatial correlations across links. The temporal module models fine-grained, per-link sequences using a SSM, while the spatial module dynamically aggregates context from other links based on learnable relevance scores. In particular, the Adaptive Attention mechanism computes similarity between the target link and its temporally shifted neighbors using a learnable similarity function, as illustrated in Equation (3). The full computation pipeline of **Tramba** is summarized in Algorithm A.3.

---

**Algorithm 3** Tramba Framework

---

**Require:** Input tensor $\mathbf{X}_{\text{in}} \in \mathbb{R}^{B \times T \times L \times F}$
**Ensure:** Prediction $\hat{\mathbf{Y}}_{\text{final}} \in \mathbb{R}^{B \times \tau \times L \times 1}$
1: $\mathbf{X}_{enc} \leftarrow \text{Linear}_{\text{embed}}(\mathbf{X}_{\text{in}}) + \text{PosEmbedding}(T)$  ▷ **Input Embedding and Encoding**
2: $\mathbf{X}_{enc} \leftarrow \text{Linear}_{\text{agg}}(\mathbf{X}_{enc})$
3: $\mathbf{X}_{enc} \leftarrow \text{Linear}_{\text{adaptive}}(\mathbf{X}_{enc})$
4: $\mathbf{X}_{seq} \leftarrow \text{reshape}(\mathbf{X}_{enc}, [B \cdot L,\ T,\ D])$  ▷ **Temporal Module: Mamba**
5: $\mathbf{h} \leftarrow \text{Mamba}(\mathbf{X}_{seq})$
6: $\mathbf{h} \leftarrow \text{LayerNorm}(\text{FeedForward}(\mathbf{h}) + \mathbf{h})$
7: $\mathbf{H}_{\text{mamba}} \leftarrow \text{reshape}(\mathbf{h}, [B,\ T,\ L,\ D])$
8: $\mathbf{H}_{\text{mamba\_last}} \leftarrow \mathbf{H}_{\text{mamba}}[:, -1, :, :]$
9: $\mathbf{H}_{\text{attn}} \leftarrow \text{AdaptiveAttention}(\mathbf{X}_{enc})$  ▷ **Spatial Module: Adaptive Attention**
10: $\mathbf{H}_{\text{attn}} \leftarrow \text{LayerNorm}(\mathbf{H}_{\text{attn}} + \mathbf{X}_{enc})$
11: $\mathbf{H}_{\text{attn\_last}} \leftarrow \mathbf{H}_{\text{attn}}[:, -1, :, :]$
12: $\mathbf{G} \leftarrow \text{nn.Parameter}(L \times 1)$  ▷ **Fusion and Multi-horizon Prediction**
13: $\boldsymbol{\omega} \leftarrow \sigma(\mathbf{G})$
14: $\mathbf{H}_{\text{fused}} \leftarrow \boldsymbol{\omega} \cdot \mathbf{H}_{\text{attn\_last}} + (1 - \boldsymbol{\omega}) \cdot \mathbf{H}_{\text{mamba\_last}}$
15: $\hat{\mathbf{Y}}_{\text{final}} \leftarrow \text{Linear}_{\tau}(\mathbf{H}_{\text{fused}})$  ▷ $(B \times L \times \tau) \rightarrow (B \times \tau \times L \times 1)$
16: **return** $\hat{\mathbf{Y}}_{\text{final}}$

---

## 4 EXPERIMENTS

### 4.1 DATASETS AND BASELINES

**Benchmarks.** We evaluate **Tramba** on three real-world traffic speed datasets, all collected from urban road networks with 5-minute aggregation intervals. METR-LA consists of 207 sensor-equipped freeway segments in Los Angeles, while PEMS-BAY covers 325 freeway segments in the San Francisco Bay Area. For Seoul, we use the TOPIS dataset (https://topis.seoul.go.kr), from which we extract 366 road links located in Gangnam-gu, one of the city's busiest districts, during June 2024. All datasets provide speed measurements, which is standard for urban probe data systems; the proposed model is general and can be applied to flow prediction as well.

**Experimental Settings.** To evaluate both short- and long-term forecasting performance, we use five prediction horizons: 1 (5 min), 6 (30 min), 12 (1 hr), 24 (2 hr), and 36 (3 hr) steps. Each input includes two normalized features per road link: lagged speed and speed change. We apply MinMax normalization to scale all input features to the [0, 1] range, which stabilizes training and prevents feature dominance. We selected final hyperparameters based on validation loss. For learning rate, we tested 1e-4, 5e-4, 1e-3; for batch size, 16, 32, 64, 128. The final configuration used a learning rate of 0.001 and a batch size of 32 for our model. For fair comparison, all models were trained with a batch size of 32. All models were optimized using Adam and trained to minimize MSE loss for up to 50 epochs with early stopping based on validation performance. All experiments were conducted on the VESSL platform using an NVIDIA A100 GPU (40 GB) and a cpu-medium instance with 2 vCPUs and 6 GB RAM, running in an Ubuntu 20.04 environment. We implemented and trained all models using PyTorch 2.5.1 with CUDA 12.1 and Python 3.12. Code is publicly available at https://github.com/tr-anon-users/tramba-code.

**Baseline Models.** We compare **Tramba** with six strong baseline models that span both attention-based and state-space modeling paradigms. These include two Transformer-based architectures—ST-Transformer Chen et al. (2022) and iTransformer Zou et al. (2024)—which are widely used in spatiotemporal forecasting due to their ability to capture long-range dependencies via attention mechanisms. In addition, we include three recent variants of Mamba Gu & Dao (2023), a state-space model optimized for efficient sequence modeling: S-Mamba Wang et al. (2025), SOR-Mamba Lee et al. (2024) and DST-Mamba He et al. (2025). These models collectively represent a diverse set of high-performing architectures and serve as strong baselines for evaluating the effectiveness of **Tramba** across a range of spatiotemporal prediction scenarios.

## 4.2 MAIN RESULTS

**Overall Performance.** **Tramba** achieves the best overall performance across nearly all horizons, datasets, and evaluation metrics. On the PEMS-BAY dataset, which is relatively clean and stable, the performance differences among models are minimal, and at the 1-step horizon **Tramba**'s MAPE of 0.12% is only marginally different (0.01–0.02 percentage points) from the best result. This small gap indicates that short-term forecasting on regular datasets presents a relatively simple task, where traffic dynamics are highly predictable and model differences become less pronounced.

In contrast, on more heterogeneous and noisy datasets such as METR-LA and TOPIS, **Tramba** consistently outperforms all baselines across every forecasting horizon and evaluation metric. The advantage becomes increasingly clear as the prediction horizon lengthens, demonstrating that **Tramba**'s design—integrating long-range temporal modeling with adaptive non-local attention—is particularly effective in capturing complex spatiotemporal dependencies under challenging and dynamic traffic conditions. Notably, the gains on TOPIS remain stable even at the 36-step horizon, further underscoring the robustness and reliability of **Tramba** in noisy real-world scenarios.

Table 1: Model performance on three traffic speed datasets (PEMS-BAY, METR-LA, and TOPIS) with input length fixed to 36 and output horizons $\{1, 6, 12, 24, 36\}$. The best results are in **bold**, and the second-best are underlined.

| Model | Horizon | PEMS-BAY | | | METR-LA | | | TOPIS | | |
|---|---|---|---|---|---|---|---|---|---|---|
| | | MAPE | MAE | MSE | MAPE | MAE | MSE | MAPE | MAE | MSE |
| ST-Transformer Chen et al. (2022) | 1-step | 0.12 | 0.03 | 0.01 | 0.78 | 0.19 | 0.09 | 0.88 | 0.20 | 0.09 |
| | 6-step | 7.48 | 2.12 | 12.46 | 8.23 | 2.31 | 13.82 | 9.63 | 2.78 | 18.56 |
| | 12-step | 9.53 | 2.66 | 18.32 | 10.18 | 2.91 | 20.39 | 11.81 | 3.35 | 27.15 |
| | 24-step | 12.95 | 3.59 | 28.37 | 14.08 | 3.88 | 31.24 | 15.77 | 4.25 | 40.85 |
| | 36-step | 16.47 | 4.42 | 39.48 | 18.52 | 4.81 | 44.07 | 21.78 | 5.56 | 64.13 |
| iTransformer Zou et al. (2024) | 1-step | 0.12 | 0.03 | 0.01 | 1.29 | 0.27 | 0.24 | 0.78 | 0.18 | 0.09 |
| | 6-step | 7.61 | 2.19 | 12.68 | 8.32 | 2.41 | 14.02 | 9.76 | 2.78 | 18.75 |
| | 12-step | 9.62 | 2.71 | 18.69 | 10.42 | 3.01 | 20.78 | 11.88 | 3.44 | 27.34 |
| | 24-step | 13.48 | 3.68 | 28.97 | 14.59 | 4.02 | 32.05 | 16.79 | 4.34 | 44.06 |
| | 36-step | 16.19 | 4.21 | 37.53 | 18.23 | 4.72 | 42.03 | 21.10 | 5.15 | 53.31 |
| Mamba Gu & Dao (2023) | 1-step | 0.12 | 0.03 | 0.01 | 0.23 | 0.06 | 0.33 | 0.27 | 0.06 | 0.36 |
| | 6-step | 8.49 | 2.31 | 12.92 | 9.01 | 2.51 | 14.23 | 12.08 | 2.86 | 18.88 |
| | 12-step | 10.09 | 2.82 | 18.52 | 10.79 | 3.11 | 20.52 | 13.31 | 3.44 | 26.75 |
| | 24-step | 13.58 | 3.49 | 27.21 | 14.71 | 3.82 | 30.02 | 16.86 | 3.93 | 37.64 |
| | 36-step | 15.02 | 4.02 | 36.49 | 17.01 | 4.39 | 40.98 | 18.36 | 4.99 | 56.39 |
| S-Mamba Wang et al. (2025) | 1-step | 0.12 | 0.03 | 0.01 | 0.47 | 0.10 | 0.02 | 0.55 | 0.11 | 0.02 |
| | 6-step | 7.88 | 2.21 | 13.08 | 8.49 | 2.41 | 14.48 | 10.10 | 2.86 | 19.21 |
| | 12-step | 9.81 | 2.79 | 18.97 | 10.52 | 3.11 | 21.03 | 12.02 | 3.44 | 27.80 |
| | 24-step | 13.09 | 3.61 | 28.49 | 14.21 | 3.89 | 31.48 | 15.70 | 4.50 | 43.54 |
| | 36-step | 15.51 | 4.09 | 36.81 | 17.52 | 4.48 | 41.52 | 19.05 | 5.15 | 56.19 |
| SOR-Mamba Lee et al. (2024) | 1-step | 0.13 | 0.04 | 0.01 | 0.21 | 0.05 | 0.01 | 0.14 | 0.03 | 0.01 |
| | 6-step | 7.42 | 2.11 | 12.72 | 8.12 | 2.29 | 14.01 | 9.42 | 2.78 | 18.75 |
| | 12-step | 9.31 | 2.72 | 18.53 | 10.09 | 2.91 | 20.69 | 12.29 | 3.35 | 27.08 |
| | 24-step | 12.48 | 3.51 | 27.92 | 13.61 | 3.82 | 30.72 | 15.02 | 4.25 | 40.52 |
| | 36-step | 15.91 | 4.21 | 35.58 | 17.79 | 4.68 | 39.81 | 19.73 | 5.48 | 52.45 |
| DST-Mamba He et al. (2025) | 1-step | **0.11** | **0.03** | **0.01** | 0.13 | 0.03 | 0.01 | 0.22 | 0.05 | 0.01 |
| | 6-step | 7.71 | 2.12 | 12.39 | 8.41 | 2.31 | 13.61 | 9.97 | 2.78 | 18.49 |
| | 12-step | 9.59 | 2.61 | 17.99 | 10.61 | 2.81 | 20.02 | 11.61 | 3.35 | 26.29 |
| | 24-step | 12.31 | 3.41 | 27.39 | 13.42 | 3.61 | 30.19 | 14.95 | 4.17 | 40.06 |
| | 36-step | 15.18 | 3.91 | 33.01 | 17.01 | 4.21 | 36.99 | 18.77 | 4.50 | 44.98 |
| Tramba (Ours) | 1-step | 0.12 | 0.03 | 0.01 | **0.12** | **0.03** | **0.01** | **0.13** | **0.03** | **0.01** |
| | 6-step | **7.01** | **2.01** | **11.91** | **7.79** | **2.19** | **13.01** | **9.08** | **2.70** | **17.90** |
| | 12-step | **9.12** | **2.49** | **17.21** | **9.91** | **2.71** | **19.12** | **11.47** | **3.19** | **25.18** |
| | 24-step | **11.99** | **3.31** | **26.99** | **12.98** | **3.49** | **29.52** | **14.68** | **4.17** | **40.32** |
| | 36-step | **14.79** | **3.79** | **32.51** | **16.51** | **4.09** | **36.49** | **17.96** | **4.17** | **44.91** |

**Ablation Analysis.** To evaluate the contribution of each component in **Tramba**, we conducted an ablation study based on 12-step forecasting on TOPIS data, as summarized in Table 2. Among all components, the removal of the adaptive attention module caused the most significant performance degradation, increasing MAPE by 3.64% (from 11.47% to 15.11%). This underscores the importance of capturing non-local road link interactions.

Table 2: Ablation for 12-step forecasting on TOPIS.

| Variant | MAPE | MAE | MSE |
|---|---|---|---|
| Tramba | 11.47 | 3.19 | 25.18 |
| w/o Attention | 15.11 | 4.19 | 37.31 |
| w/o Similarity | 14.35 | 3.82 | 32.37 |
| w/o Gate Fusion | 14.01 | 3.54 | 29.73 |
| w/o Embedding | 13.22 | 3.27 | 26.51 |

Similarly, replacing the learnable similarity function with a simple dot-product alternative resulted in worse performance (MAPE 14.35%, MSE 32.37 $(km/h)^2$), clearly demonstrating the advantage of data-driven and adaptive relevance estimation between spatial entities in complex traffic networks. Other components showed smaller, yet measurable impacts. Removing the learnable gate fusion increased MAPE to 14.01%, confirming that **Tramba**'s integration is not trivial stacking of modules but rather a necessary mechanism to effectively balance complementary predictive signals. Lastly, removing the adaptive embedding layer raised the error to 13.22%, further suggesting its auxiliary role in capturing input heterogeneity across diverse traffic conditions. Overall, these results confirm that **Tramba**'s fusion mechanism effectively addresses key challenges in spatiotemporal forecasting by balancing long-term memory and non-local contextual information.

**Confidence Analysis.** To evaluate both the predictive accuracy and robustness of our model, we conducted a comprehensive confidence interval (CI) analysis based on ten independent training runs for **Tramba** as well as all baseline models (see Appendix E, Table E.1 for the full CI results). For each run, we recorded the evaluation metrics on the test set and computed the 95% confidence intervals to quantify the variability due to random initialization. The results reveal that all models exhibit relatively narrow CIs, suggesting that their performance is not highly sensitive to initialization and training noise. However, notable differences in consistency and absolute performance still emerge across settings. In particular, **Tramba** consistently outperformed the baseline models across all evaluation metrics—achieving the lowest mean absolute percentage error (MAPE) of 11.47 ± 0.19%, the lowest mean absolute error (MAE) of 3.19 ± 0.23 km/h, and the lowest mean squared error (MSE) of 25.18 ± 1.12 $(km/h)^2$. **Tramba** not only achieved the best average performance but also showed the smallest variation across runs, underscoring its stability and reliability under stochastic training.

## 4.3 ATTENTION ANALYSIS

**Localized Trends.** Figure 2 shows the result of **Tramba**, which forecasts future speeds for a target link (Link 156) by retrieving relevant historical patterns from spatially distant but temporally aligned road network segments. The upper plot shows the detailed temporal speed profile of the target link over a full day. The figure further highlights three non-adjacent links that were deliberately selected by the model despite not being physically connected to the target. These examples demonstrate that **Tramba** does not rely solely on recent values or adjacent segments, but instead attends to distant links that exhibit temporal patterns similar to the target, thereby improving prediction accuracy.

- **Link 101 (blue)** shows a steady pattern during the afternoon (12:00–19:00), resembling stable traffic flow on the target link from morning to early afternoon.
- **Link 240 (orange)** shows congestion and recovery patterns during the late afternoon to evening (16:00–22:00), similar to the target link's pattern during that period.
- **Link 183 (red)** shows a transition from congestion to relief in the afternoon to evening (13:00–18:00), which aligns with the target link's evening recovery phase.

By integrating non-local attention with a memory-preserving temporal encoder, **Tramba** not only improves predictive accuracy but also offers two theoretical advantages: *robustness*, since the mechanism is stable to index permutations and temporal shifts, and *interpretability*, as the attention weights reveal which distant links and alignments contribute most to the forecast. This joint design achieves a principled balance between temporal continuity and spatial similarity, underscoring the novelty of Tramba's dynamic integration and its ability to deliver both resilience to noise and transparent insights into urban traffic dynamics.

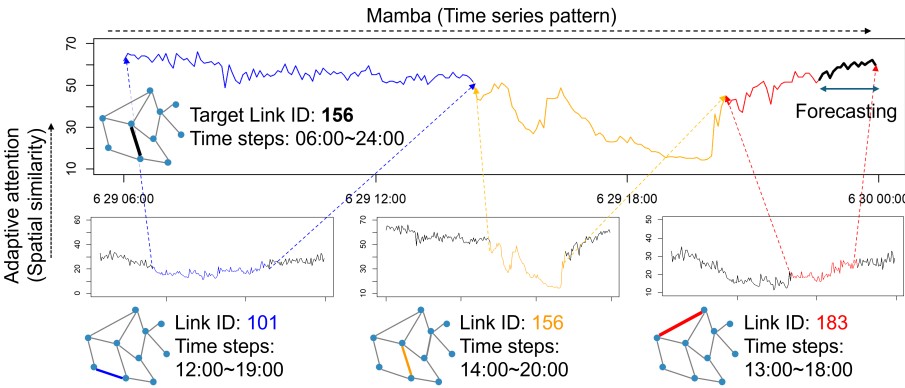

Figure 2: Attention based on temporal similarity. **Tramba** retrieves spatially non-adjacent links whose speed patterns align with different time segments of the target link (Link 156), effectively leveraging temporally coherent signals from distant locations to capture long-range temporal dependencies and enhance forecasting accuracy across the network.

**Temporal Patterns.** Figure 3a compares the traffic speed patterns of the target link (Link 156), its top-attended link, and a physically connected downstream link over the period from 6:00 to 24:00. Although the downstream link is geographically close, its temporal pattern deviates markedly from the target—particularly during the onset and recovery phases of congestion. In contrast, the top-attended link, despite being spatially distant, exhibits a highly synchronized trajectory with the target. This suggests that **Tramba** assigns attention based on dynamic similarity rather than physical proximity. These results demonstrate **Tramba**'s ability to adaptively capture both local and non-local dependencies by focusing on temporal alignment. Such flexibility is especially beneficial in modeling complex traffic behavior in heterogeneous urban networks.

**Non-local Influence.** To interpret these results, we analyze spatial attention scores originating from a specific road link, Link 156. As shown in Figure 3b, we visualize the attention weights assigned by Link 156 to several other links: the top-3 most attended links (ranked by attention score) and two physically connected links, upstream (Up) and downstream (Down) links. Surprisingly, the top-1 to top-3 links attended by **Tramba** are spatially distant from Link 156, located in entirely different areas of the network. Despite this lack of physical proximity, these links receive higher attention scores (the lower left quadrant of the figure). In contrast, the physically connected upstream and downstream links receive low attention weights (highlighted in red text). These results illustrate that mere connectivity does not guarantee predictive relevance. Instead, **Tramba** prioritizes links that share temporal dynamics with the target, effectively learning non-local but highly informative dependencies.

**Global Network Trends.** Figure 3c illustrates the spatial distribution of high-attention links across the network as learned by **Tramba**. The color of each link reflects its spatial distance from the top-1 most attended link, revealing how **Tramba** allocates attention based not only on proximity but also on temporal correlation. Red links indicate physically connected upstream or downstream links; however, among all 366 links, only 10 exhibit dominant attention toward directly connected links, suggesting that physical adjacency accounts for a limited portion of influential relationships. The remaining links, which are not physically connected to the target, are categorized by distance: yellow to green lines represent nearby but unconnected links, while blue to purple lines denote distant links that nonetheless receive high attention scores. The results reveal that downtown links tend to share similar speed patterns with nearby links due to consistent congestion, high signal density, and low speed variability. In contrast, upstream arterial roads often align more closely with distant links, as they experience fewer signals, greater speed fluctuations, and more dynamic transitions.

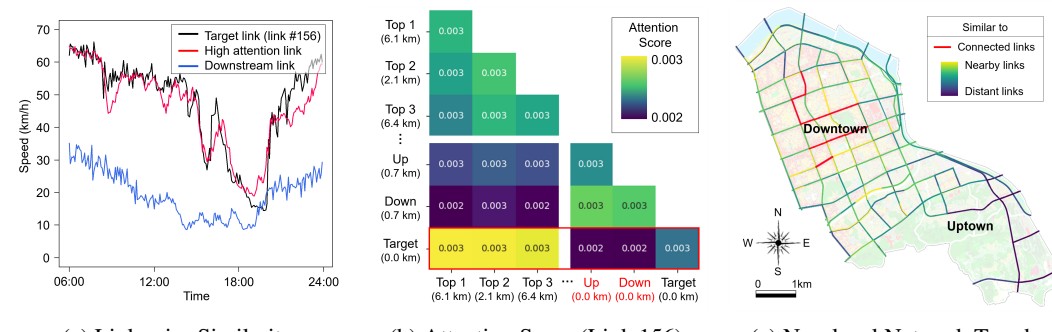

(a) Link-wise Similarity.  (b) Attention Score (Link 156).  (c) Non-local Network Trends.

Figure 3: Visual interpretation of **Tramba**'s attention behavior. (a) Temporal comparison of target, attended, and downstream links, showing temporal similarity over physical proximity. (b) Attention scores for Link 156, where distant links dominate over physically connected ones. (c) Network-wide distribution of attended links, revealing non-local but temporally aligned dependencies.

## 5 CONCLUSIONS

**Conclusion.** We introduced **Tramba**, a time-series forecasting model that integrates Mamba-based long-range temporal modeling with adaptive spatial attention. **Tramba** captures dynamic, non-local traffic dependencies without relying on predefined topologies, and achieves consistent improvements over strong baselines across multiple horizons. Our results suggest that **Tramba** effectively models the complex and irregular nature of urban traffic, mitigating time-shift misalignments in temporal dependencies while maintaining strong generalization across scenarios.

**Limitations.** While **Tramba** effectively captures complex spatiotemporal dynamics, several limitations remain. First, the adaptive attention mechanism is fully data-driven and lacks causal constraints or shift-aware priors, which could improve interpretability. Second, the current framework does not incorporate intersection-level features or traffic signal control—factors especially influential in urban environments. Lastly, scalability may be a concern for larger networks or longer historical sequences, suggesting the need for memory-efficient approximations.

**Broader Impact.** Our model, **Tramba**, provides a flexible and generalizable framework for modeling spatiotemporal dynamics in traffic systems without relying on fixed spatial graphs. Its ability to uncover implicit functional relationships among distant links opens up new opportunities for improving traffic forecasting, incident detection, and infrastructure planning. Beyond traffic, **Tramba** provides a transferable blueprint for other spatiotemporal systems, including mobility-on-demand services and sensor networks. This demonstrates that **Tramba** is not merely a hybrid but a general framework for irregular dependencies.

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

# A ALGORITHMIC DETAILS OF TRAMBA

This section describes the core computational pipeline of **Tramba**, starting from the raw input and detailing the operations in the temporal and spatial modules, including key parameter roles.

## A.1 INPUT EMBEDDING

**Indices and dimensions.** Batch $B$, time $T$, links $L$, raw features $F$, hidden size $D$, prediction horizon $\tau$.

We define the spatiotemporal input as a 4D tensor

$$\mathbf{X}_{\text{in}} \in \mathbb{R}^{B \times T \times L \times F}, \tag{A.1}$$

where $B$ is the batch size, $T$ is the number of historical steps, $L$ is the number of spatial entities (e.g., links or sensors), and $F$ is the number of input features per location and time.

**Shared linear embedding and positional term.** A shared linear layer projects the raw inputs to a hidden dimension $D$. To inject temporal order, a learnable positional embedding $\mathbf{P} \in \mathbb{R}^{T \times D}$ is added along the time axis:

$$\mathbf{Z}^{(0)} = \texttt{Linear}_{\text{embed}}(\mathbf{X}_{\text{in}}) \in \mathbb{R}^{B \times T \times L \times D}, \tag{A.2}$$

$$\mathbf{Z}^{(1)}_{b,t,i,:} = \mathbf{Z}^{(0)}_{b,t,i,:} + \mathbf{P}_{t,:}. \tag{A.3}$$

Here, $\mathbf{P}$ is broadcast across the batch and link dimensions.

**Feature aggregation / dimensional alignment.** An additional linear layer mixes features within the hidden channel and can align dimensions for downstream modules:

$$\mathbf{X}_{\text{enc}} = \texttt{Linear}_{\text{agg}}(\mathbf{Z}^{(1)}) \in \mathbb{R}^{B \times T \times L \times D}. \tag{A.4}$$

**Notes.** (i) If the downstream temporal and spatial modules use different hidden sizes, set $\texttt{Linear}_{\text{agg}} : D \to D'$ and denote the resulting size by $D'$. (ii) The encoded tensor $\mathbf{X}_{\text{enc}}$ is the common input to the Mamba block and the adaptive attention module.

## A.2 MAMBA BLOCK

**Selective State Space Modeling.** To capture long-range temporal dependencies per link, **Tramba** applies a Mamba block to the encoded inputs $\mathbf{X}_{\text{enc}} \in \mathbb{R}^{B \times T \times L \times D}$. All linear layers below are shared across links and applied pointwise over $(b, t, i)$.

**Dual-stream projection.** Split the input into an SSM-driving stream $\tilde{\mathbf{x}}$ and a gating stream $\mathbf{z}$:

$$\tilde{\mathbf{x}}, \mathbf{z} = \texttt{Linear}_{\text{split}}(\mathbf{X}_{\text{enc}}), \qquad \tilde{\mathbf{x}}, \mathbf{z} \in \mathbb{R}^{B \times T \times L \times D}. \tag{A.5}$$

**Local temporal preprocessing.** Extract short-range patterns with a depthwise 1D convolution along time, followed by SiLU:

$$\mathbf{x}' = \texttt{SiLU}\big(\texttt{Conv1D}(\tilde{\mathbf{x}})\big), \qquad \mathbf{x}' \in \mathbb{R}^{B \times T \times L \times D}. \tag{A.6}$$

**Data-dependent SSM parameters.** Project $\mathbf{x}'$ to continuous-time SSM parameters:

$$\mathbf{A} = \texttt{Linear}_A(\mathbf{x}'), \quad \mathbf{B} = \texttt{Linear}_B(\mathbf{x}'), \quad \mathbf{C} = \texttt{Linear}_C(\mathbf{x}'), \tag{A.7}$$

and compute a positive step size via Softplus:

$$\boldsymbol{\Delta} = \texttt{Softplus}\big(\boldsymbol{\theta} + \texttt{Linear}_\Delta(\mathbf{x}')\big), \qquad \boldsymbol{\theta} \in \mathbb{R}^D. \tag{A.8}$$

**Discretization.** Discretize $(\mathbf{A}, \mathbf{B})$ with step $\boldsymbol{\Delta}$ (e.g., bilinear or zero-order hold):

$$\bar{\mathbf{A}}, \bar{\mathbf{B}} = \texttt{discretize}(\boldsymbol{\Delta}, \mathbf{A}, \mathbf{B}). \tag{A.9}$$

**Selective SSM scan.** For each $(b, i)$ sequence and $t = 1, \ldots, T$,

$$\mathbf{y}_t = \bar{\mathbf{A}}_t \, \mathbf{y}_{t-1} + \bar{\mathbf{B}}_t \, \mathbf{x}'_t, \tag{A.10}$$

$$\hat{\mathbf{y}}_t = \mathbf{C}_t \, \mathbf{y}_t. \tag{A.11}$$

where all symbols are $D$-dimensional per $(b, t, i)$ and the scan runs in parallel across $B$ and $L$.

**Gating and projection.** Modulate the SSM output by the auxiliary stream and project to the latent space:

$$\mathbf{y}'_t = \hat{\mathbf{y}}_t \odot \mathtt{SiLU}(\mathbf{z}_t), \tag{A.12}$$

$$\mathbf{h}_t = \mathtt{Linear}_{\text{out}}(\mathbf{y}'_t). \tag{A.13}$$

**Residual refinement.** Apply a position-wise feedforward block with residual:

$$\mathbf{o}_t = \mathtt{FFN}(\mathbf{h}_t) + \mathbf{h}_t. \tag{A.14}$$

**Output shape.** Stacking over $t$ and restoring dimensions yields

$$\mathbf{H}_{\text{mamba}} = \mathtt{reshape}\big(\{\mathbf{o}_t\}_{t=1}^{T}, \, [B, T, L, D]\big), \qquad \mathbf{H}_{\text{mamba\_last}} = \mathbf{H}_{\text{mamba}}[:, T, :, :] \in \mathbb{R}^{B \times L \times D}. \tag{A.15}$$

**Remarks.** (i) $\mathtt{Linear}_{\text{split}}$ typically outputs $2D$ channels that are partitioned into $\tilde{\mathbf{x}}$ and $\mathbf{z}$. (ii) $\mathtt{Conv1D}$ is causal or same-padded along time; kernel size $k$ is a hyperparameter. (iii) The scan in equation A.10 is implemented with the fused selective-SSM kernel for linear-time complexity $\mathcal{O}(B\,L\,T\,D)$.

**Feedforward and Normalization.** The Mamba output is refined by a position-wise feedforward network with a residual connection, then normalized. For all $(b, t, i)$,

$$\mathbf{h}_{b,t,i,:} \leftarrow \mathtt{FFN}\big(\mathbf{h}_{b,t,i,:}\big) + \mathbf{h}_{b,t,i,:}, \qquad \mathbf{H}_{\text{mamba}} \leftarrow \mathtt{LayerNorm}\big(\mathbf{h}\big), \tag{A.16}$$

where $\mathtt{FFN}(x) = \mathtt{Linear}_2(\mathtt{ReLU}(\mathtt{Linear}_1(x)))$ with $\mathtt{Linear}_1 : \mathbb{R}^D \to \mathbb{R}^{2D}$ and $\mathtt{Linear}_2 : \mathbb{R}^{2D} \to \mathbb{R}^D$. The final tensor preserves shape $\mathbf{H}_{\text{mamba}} \in \mathbb{R}^{B \times T \times L \times D}$.

*Remark on horizons.* This block produces per-time features up to the last observed step ($T$). It is *not* multi-horizon decoding. The multi-horizon prediction ($\tau$ steps) is performed later by the head $\mathtt{Linear}_\tau$ applied to the fused representation (see fusion section).

A.3   ADAPTIVE ATTENTION

Let $\mathbf{X}_{\text{enc}} \in \mathbb{R}^{B \times T \times L \times D}$ be the encoded input. Given a set of temporal shifts $\mathcal{S} = \{-s_1, \ldots, -1, 0\} \cup \mathtt{extra\_shifts}$ with $|\mathcal{S}| = S$, we construct shifted tensors by rolling along time with padding and validity masks:

$$\mathbf{X}^{(s)} = \mathtt{Shift}(\mathbf{X}_{\text{enc}}, s), \qquad s \in \mathcal{S}, \tag{A.17}$$

---

**Algorithm A.1** Forward Computation of the Mamba Block (per-link temporal encoder)

---

**Require:** Encoded inputs reshaped to sequences $\mathbf{X}_{\text{seq}} \in \mathbb{R}^{(B \cdot L) \times T \times D}$
**Ensure:** Temporal features $\mathbf{H}_{\text{mamba}} \in \mathbb{R}^{B \times T \times L \times D}$
 1: $\tilde{\mathbf{x}}, \mathbf{z} \leftarrow \mathtt{Linear}_{\text{split}}(\mathbf{X}_{\text{seq}})$ ▷ split into state/gate streams
 2: $\mathbf{x}' \leftarrow \mathtt{SiLU}(\mathtt{Conv1D}(\tilde{\mathbf{x}}))$ ▷ temporal filtering
 3: $\mathbf{A} \leftarrow \mathtt{Linear}_A(\mathbf{x}')$; $\quad \mathbf{B} \leftarrow \mathtt{Linear}_B(\mathbf{x}')$; $\quad \mathbf{C} \leftarrow \mathtt{Linear}_C(\mathbf{x}')$
 4: $\boldsymbol{\Delta} \leftarrow \mathtt{Softplus}\big(\boldsymbol{\theta} + \mathtt{Linear}_\Delta(\mathbf{x}')\big)$
 5: $\bar{\mathbf{A}}, \bar{\mathbf{B}} \leftarrow \mathtt{discretize}(\boldsymbol{\Delta}, \mathbf{A}, \mathbf{B})$
 6: $\mathbf{y} \leftarrow \mathtt{SelectiveSSM}(\bar{\mathbf{A}}, \bar{\mathbf{B}}, \mathbf{C})(\mathbf{x}')$ ▷ scan over $t$ (parallel across $B \cdot L$)
 7: $\mathbf{y}' \leftarrow \mathbf{y} \odot \mathtt{SiLU}(\mathbf{z})$ ▷ gated modulation
 8: $\mathbf{h} \leftarrow \mathtt{Linear}_{\text{out}}(\mathbf{y}')$
 9: $\mathbf{h} \leftarrow \mathtt{FFN}(\mathbf{h}) + \mathbf{h}$; $\quad \mathbf{h} \leftarrow \mathtt{LayerNorm}(\mathbf{h})$
10: $\mathbf{H}_{\text{mamba}} \leftarrow \mathtt{reshape}(\mathbf{h}, [B, T, L, D])$
11: **return** $\mathbf{H}_{\text{mamba}}$

---

The operator $\texttt{Shift}(\cdot, s)$ aligns potential causes in the past with the target time index, enabling lag-aware comparisons (e.g., upstream-to-downstream delays). Padding is applied where the shift crosses sequence boundaries, and a mask later suppresses these invalid entries.

and stack them along a new shift axis:

$$\mathbf{X}_{\text{shifted}} = \texttt{stack}\big(\{\mathbf{X}^{(s)}\}_{s \in \mathcal{S}}, \text{axis} = 1\big) \in \mathbb{R}^{B \times S \times T \times L \times D}. \tag{A.18}$$

Stacking introduces an explicit "shift" dimension that we will normalize over together with the source-link axis. This layout allows efficient batched scoring across all $(s, j)$ pairs while preserving time and feature axes.

**Learnable similarity.** For a target link $i$ and a source link $j$ at shift $s$, define the similarity

$$S_{b,s,t,i,j} = \texttt{Sim}_\theta\Big(\mathbf{X}_{\text{enc}, b,t,i,:}, \ \mathbf{X}^{(s)}_{b,t,j,:}\Big), \tag{A.19}$$

$\texttt{Sim}_\theta$ can instantiate a scaled dot product or an MLP over cross-features, trading compute for expressivity. The score compares the target's current state with each source's time-shifted state, capturing non-local, delay-aware correlations.

We average scores over time (keeping spatial and shift axes):

$$\bar{S}_{b,s,i,j} = \frac{1}{T} \sum_{t=1}^{T} S_{b,s,t,i,j}. \tag{A.20}$$

Time-averaging stabilizes weights and reduces variance from short-lived fluctuations. It also removes the time axis before normalization, simplifying the subsequent softmax over $(s, j)$.

With a mask $M_{b,s,i,j} \in \{0, -\infty\}$ for invalid (padded) positions, we normalize **jointly over** $(s, j)$:

$$A^{(s)}_{b,i,j} = \frac{\exp(\bar{S}_{b,s,i,j} + M_{b,s,i,j})}{\sum\limits_{j'=1}^{L} \sum\limits_{s' \in \mathcal{S}} \exp(\bar{S}_{b,s',i,j'} + M_{b,s',i,j'})}. \tag{A.21}$$

Joint normalization lets each target distribute probability mass across all sources at all shifts, naturally selecting both the link and its effective lag. The mask adds $-\infty$ to padded entries to ensure they receive zero probability.

**Attention output.** Align values and weights and aggregate along the flattened $(s, j)$ axis:

$$\mathbf{V} = \texttt{reshape}(\mathbf{X}_{\text{shifted}}, \ [B, \ T, \ S{\cdot}L, \ D]), \tag{A.22}$$

$$\mathbf{A} = \texttt{reshape}\Big(\{A^{(s)}_{b,i,j}\}, \ [B, \ L, \ S{\cdot}L]\Big), \tag{A.23}$$

$$\mathbf{H}_{\text{attn}, b,:,i,:} = \sum_{u=1}^{S{\cdot}L} \mathbf{A}_{b,i,u} \, \mathbf{V}_{b,:,u,:}, \quad \Rightarrow \quad \mathbf{H}_{\text{attn}} \in \mathbb{R}^{B \times T \times L \times D}. \tag{A.24}$$

We flatten the $(s, j)$ axes to enable a single batched weighted sum per target link. The result preserves the original time and feature dimensions, yielding a temporally aligned spatial context for each link.

Finally, we apply a residual connection with the encoded input and layer normalization:

$$\mathbf{H}_{\text{attn}} \leftarrow \texttt{LayerNorm}\big(\mathbf{H}_{\text{attn}} + \mathbf{X}_{\text{enc}}\big). \tag{A.25}$$

The residual path retains local information present in $\mathbf{X}_{\text{enc}}$, while LayerNorm improves optimization stability. This output is then compatible in shape and scale with the temporal branch for downstream fusion.

## A.4 FUSION AND OUTPUT

After obtaining two complementary representations—$\mathbf{H}_{\text{attn}}$ from the adaptive attention branch and $\mathbf{H}_{\text{mamba}}$ from the temporal branch—**Tramba** fuses them with a learnable link-wise gate. We use the last observed time slice from each branch (consistent with Algorithm A.3):

$$\mathbf{H}_{\text{attn\_last}} = \mathbf{H}_{\text{attn}}[:, T{-}1, :, :], \qquad \mathbf{H}_{\text{mamba\_last}} = \mathbf{H}_{\text{mamba}}[:, T{-}1, :, :] \ \in \ \mathbb{R}^{B \times L \times D}. \tag{A.26}$$

---

**Algorithm A.2** Adaptive Attention (inputs/outputs aligned with the main framework)

---

**Require:** Encoded input $\mathbf{X}_{\text{enc}} \in \mathbb{R}^{B \times T \times L \times D}$, shift set $\mathcal{S}$
**Ensure:** $\mathbf{H}_{\text{attn}} \in \mathbb{R}^{B \times T \times L \times D}$
 1: For each $s \in \mathcal{S}$: $\mathbf{X}^{(s)} \leftarrow \texttt{Shift}(\mathbf{X}_{\text{enc}}, s)$ with mask $M^{(s)}$
 2: $\mathbf{X}_{\text{shifted}} \leftarrow \texttt{stack}\big(\{\mathbf{X}^{(s)}\}_{s \in \mathcal{S}}, \text{axis} = 1\big)$ $\qquad\qquad\qquad$ $\triangleright [B, S, T, L, D]$
 3: $\mathbf{S} \leftarrow \texttt{Sim}_\theta\big(\mathbf{X}_{\text{enc}}\,(\mathbf{Q}),\ \mathbf{X}_{\text{shifted}}\,(\mathbf{K})\big)$ $\qquad\qquad\qquad$ $\triangleright [B, S, T, L, L]$
 4: $\bar{\mathbf{S}} \leftarrow \texttt{Mean}(\mathbf{S}, \text{dim} = 2)$; $\bar{\mathbf{S}} \leftarrow \bar{\mathbf{S}} + M$ $\qquad\qquad$ $\triangleright$ time avg + mask
 5: $\mathbf{A} \leftarrow \texttt{softmax}\big(\texttt{reshape}(\bar{\mathbf{S}}), \text{dim} = 2\big)$ $\qquad\quad$ $\triangleright$ over $(s, j) \Rightarrow [B, L, S \cdot L]$
 6: $\mathbf{V} \leftarrow \texttt{reshape}(\mathbf{X}_{\text{shifted}}, [B, T, S \cdot L, D])$
 7: For $i = 1 \dots L$: $\mathbf{H}_{\text{attn},:,i,:} \leftarrow \mathbf{A}_{:,i,:}\,\mathbf{V}$ $\qquad\qquad$ $\triangleright$ batched weighted sum over $S \cdot L$
 8: $\mathbf{H}_{\text{attn}} \leftarrow \texttt{LayerNorm}(\mathbf{H}_{\text{attn}} + \mathbf{X}_{\text{enc}})$
 9: **return** $\mathbf{H}_{\text{attn}}$

---

Taking the last slice summarizes (i) non-local spatial context aligned to the most recent observation and (ii) per-link temporal state. Using the same cut for both branches keeps shapes aligned and reduces compute at the fusion stage.

**Gate parameterization.** We learn a link-wise logit $\mathbf{G} \in \mathbb{R}^{L \times 1}$ and map it through a sigmoid; the result is broadcast over batch and channel:

$$\boldsymbol{\omega} = \sigma(\mathbf{G}) \quad \Rightarrow \quad \boldsymbol{\omega} \in \mathbb{R}^{B \times L \times D} \text{ (via broadcasting).} \tag{A.27}$$

This design lets each link choose its own spatial–temporal balance while keeping the fusion light-weight (no extra MLP over $B$ or $T$). We initialize $\mathbf{G}_\ell = 0 \Rightarrow \omega_\ell = 0.5$ to avoid early saturation and give both branches equal influence at the start of training.

**Fusion.** The fused representation is a convex combination of the two last-step embeddings:

$$\mathbf{H}_{\text{fused}} = \boldsymbol{\omega} \odot \mathbf{H}_{\text{attn\_last}} + \big(1 - \boldsymbol{\omega}\big) \odot \mathbf{H}_{\text{mamba\_last}} \in \mathbb{R}^{B \times L \times D}. \tag{A.28}$$

When $\omega_\ell \approx 1$, link $\ell$ relies more on non-local similarity; when $\omega_\ell \approx 0$, it relies on temporal continuity. Broadcasting preserves the per-link structure and makes the operation embarrassingly parallel over $B$ and $L$.

**Multi-horizon head.** We decode $\tau$ future steps directly with a shared linear head:

$$\hat{\mathbf{Y}}_{1:\tau} = \texttt{Linear}_\tau(\mathbf{H}_{\text{fused}}) \in \mathbb{R}^{B \times \tau \times L \times 1}. \tag{A.29}$$

$\texttt{Linear}_\tau$ maps $D \to \tau$ (implementable as a $1 \times 1$ conv over channels) and is reshaped to $[B, \tau, L, 1]$. Unlike repetition, this produces distinct horizon-specific values and enables horizon-wise calibration (e.g., larger uncertainty at longer $\tau$).

**Notes.** (i) In single-step settings, set $\tau = 1$; then $\texttt{Linear}_\tau$ reduces to a scalar head per $(b, i)$. (ii) For stability, we optionally clip gate logits (e.g., $\mathbf{G} \in [-5, 5]$) and exclude $\mathbf{G}$ from weight decay to prevent biasing the balance. (iii) The prediction tensor is already shaped for horizon-wise metrics (MAPE@1/@6/@12/@24/@36) and for per-horizon loss weighting if desired. (iv) A time-dependent gate (function of $\mathbf{H}_{\text{attn}}, \mathbf{H}_{\text{mamba}}$) is a drop-in alternative but was not needed empirically; the link-wise gate achieved a better accuracy/complexity trade-off.

A.5 TRAMBA: SUMMARY AND EXECUTION FLOW

We consider the spatiotemporal input tensor

$$\mathbf{X}_{\text{in}} \in \mathbb{R}^{B \times T \times L \times F}, \tag{A.30}$$

where $B$ is the batch size, $T$ the number of historical time steps, $L$ the number of links (sensors), and $F$ the number of raw features per link and time (e.g., speed, speed change). The model produces multi-horizon forecasts

$$\hat{\mathbf{Y}}_{1:\tau} \in \mathbb{R}^{B \times \tau \times L \times 1}, \tag{A.31}$$

for a prediction horizon of length $\tau$.

**(1) Input embedding and positional encoding.** Raw inputs are projected into a hidden space of size $D$ via a shared linear layer and augmented with learnable temporal positions:

$$\mathbf{Z}^{(0)} = \texttt{Linear}_{\text{embed}}(\mathbf{X}_{\text{in}}) \in \mathbb{R}^{B \times T \times L \times D}, \tag{A.32}$$

$$\mathbf{Z}^{(1)}_{b,t,\ell,:} = \mathbf{Z}^{(0)}_{b,t,\ell,:} + \mathbf{P}_{t,:}, \qquad \mathbf{P} \in \mathbb{R}^{T \times D}. \tag{A.33}$$

*Rationale.* equation A.32 is a pointwise channel projection shared across links and time; equation A.33 injects temporal order so that identical values at different time indices remain distinguishable.

**(2) Feature adaptation.** Two additional pointwise projections mix features and align dimensions for downstream modules:

$$\mathbf{X}_{\text{enc}} = \texttt{Linear}_{\text{agg}}(\mathbf{Z}^{(1)}), \tag{A.34}$$

$$\mathbf{X}_{\text{enc}} = \texttt{Linear}_{\text{adaptive}}(\mathbf{X}_{\text{enc}}) \in \mathbb{R}^{B \times T \times L \times D}. \tag{A.35}$$

*Rationale.* $\texttt{Linear}_{\text{agg}}$ performs feature mixing; $\texttt{Linear}_{\text{adaptive}}$ ensures the hidden width matches both temporal and spatial branches.

**(3) Temporal module: Mamba.** We reshape per link, run the Mamba block, refine with a residual FFN, and restore the spatiotemporal layout:

$$\mathbf{X}_{\text{seq}} = \texttt{reshape}(\mathbf{X}_{\text{enc}}, [B \cdot L,\ T,\ D]), \tag{A.36}$$

$$\mathbf{h} = \texttt{Mamba}(\mathbf{X}_{\text{seq}}) \in \mathbb{R}^{B \cdot L \times T \times D}, \tag{A.37}$$

$$\mathbf{h} = \texttt{LayerNorm}\big(\texttt{FeedForward}(\mathbf{h}) + \mathbf{h}\big), \tag{A.38}$$

$$\mathbf{H}_{\text{mamba}} = \texttt{reshape}(\mathbf{h}, [B,T,L,D]), \qquad \mathbf{H}_{\text{mamba\_last}} = \mathbf{H}_{\text{mamba}}[:, T-1, :, :] \in \mathbb{R}^{B \times L \times D}. \tag{A.39}$$

*Rationale.* equation A.36–equation A.39 model long-range temporal dependencies *per link* with no spatial leakage; the last slice summarizes the most recent temporal state for fusion.

**(4) Spatial module: Adaptive attention.** We compute temporally aware cross-link context from the encoded inputs (not from the reshaped sequence), add a residual, and normalize:

$$\mathbf{H}_{\text{attn}} = \texttt{AdaptiveAttention}(\mathbf{X}_{\text{enc}}) \in \mathbb{R}^{B \times T \times L \times D}, \tag{A.40}$$

$$\mathbf{H}_{\text{attn}} = \texttt{LayerNorm}\big(\mathbf{H}_{\text{attn}} + \mathbf{X}_{\text{enc}}\big), \quad \mathbf{H}_{\text{attn\_last}} = \mathbf{H}_{\text{attn}}[:, T-1, :, :] \in \mathbb{R}^{B \times L \times D}. \tag{A.41}$$

*Rationale.* The residual path preserves local information while the attention branch aggregates non-local, time-aligned signals from other links.

**(5) Fusion and multi-horizon head.** A link-wise gate balances spatial and temporal evidence. Let $\mathbf{G} \in \mathbb{R}^{L \times 1}$ and $\boldsymbol{\omega} = \sigma(\mathbf{G})$, broadcast over batch and channel:

$$\boldsymbol{\omega} \xrightarrow{\text{broadcast}} \mathbb{R}^{B \times L \times D}. \tag{A.42}$$

The fused representation and the direct multi-horizon head are

$$\mathbf{H}_{\text{fused}} = \boldsymbol{\omega} \odot \mathbf{H}_{\text{attn\_last}} + \big(1 - \boldsymbol{\omega}\big) \odot \mathbf{H}_{\text{mamba\_last}} \in \mathbb{R}^{B \times L \times D}, \tag{A.43}$$

$$\hat{\mathbf{Y}}_{1:\tau} = \texttt{Linear}_\tau\big(\mathbf{H}_{\text{fused}}\big) \in \mathbb{R}^{B \times \tau \times L \times 1}. \tag{A.44}$$

*Rationale.* $\texttt{Linear}_\tau$ predicts *distinct* targets for each future step; no output duplication is performed. Initializing $\mathbf{G}_\ell = 0$ yields $\omega_\ell = 0.5$, giving both branches equal weight at training start.

**(6) Objective and evaluation.** We minimize a convex combination of MAE and MSE over all horizons:

$$\mathcal{L} = \lambda_1 \, \text{MAE}\big(\hat{\mathbf{Y}}_{1:\tau}, \mathbf{Y}_{1:\tau}\big) + \lambda_2 \, \text{MSE}\big(\hat{\mathbf{Y}}_{1:\tau}, \mathbf{Y}_{1:\tau}\big), \tag{A.45}$$

and report horizon-wise metrics such as MAPE@1, @6, @12, @24, @36.

**(7) Computational profile.** The temporal branch (Equation (A.36)–equation A.39) runs in $\mathcal{O}(B\,L\,T\,D)$. The attention branch scales as $\mathcal{O}(B\,T\,L^2\,S\,D)$ with $S$ time shifts (aggregation has the same order). Memory is dominated by storing $\mathbf{X}_{\text{enc}}$ and attention logits over $(S, L, L)$.

**Interpretation.** The last-step encodings in Equations (A.39) and (A.41) summarize recent temporal evolution and non-local spatial context, respectively. The gate in Equation (A.43) adapts per link between temporal continuity and non-local similarity, and the head in Equation (A.44) decodes the fused state into horizon-specific forecasts.

---

**Algorithm A.3** Tramba Framework

---

**Require:** $\mathbf{X}_{\text{in}} \in \mathbb{R}^{B \times T \times L \times F}$
**Ensure:** $\hat{\mathbf{Y}}_{\text{final}} \in \mathbb{R}^{B \times \tau \times L \times 1}$
1: $\mathbf{X}_{\text{enc}} \leftarrow \texttt{Linear}_{\text{embed}}(\mathbf{X}_{\text{in}}) + \texttt{PosEmbedding}(T)$         ▷ embed & time positions
2: $\mathbf{X}_{\text{enc}} \leftarrow \texttt{Linear}_{\text{agg}}(\mathbf{X}_{\text{enc}}); \quad \mathbf{X}_{\text{enc}} \leftarrow \texttt{Linear}_{\text{adaptive}}(\mathbf{X}_{\text{enc}})$
3: $\mathbf{X}_{\text{seq}} \leftarrow \texttt{reshape}(\mathbf{X}_{\text{enc}}, [B{\cdot}L, T, D])$         ▷ **Temporal: Mamba**
4: $\mathbf{h} \leftarrow \texttt{Mamba}(\mathbf{X}_{\text{seq}})$
5: $\mathbf{h} \leftarrow \texttt{LayerNorm}(\texttt{FeedForward}(\mathbf{h}) + \mathbf{h})$
6: $\mathbf{H}_{\text{mamba}} \leftarrow \texttt{reshape}(\mathbf{h}, [B, T, L, D]); \quad \mathbf{H}_{\text{mamba\_last}} \leftarrow \mathbf{H}_{\text{mamba}}[:, T-1, :, :]$
7: $\mathbf{H}_{\text{attn}} \leftarrow \texttt{AdaptiveAttention}(\mathbf{X}_{\text{enc}})$         ▷ **Spatial: Adaptive Attention**
8: $\mathbf{H}_{\text{attn}} \leftarrow \texttt{LayerNorm}(\mathbf{H}_{\text{attn}} + \mathbf{X}_{\text{enc}}); \quad \mathbf{H}_{\text{attn\_last}} \leftarrow \mathbf{H}_{\text{attn}}[:, T-1, :, :]$
9: $\mathbf{G} \leftarrow \texttt{Parameter}(L, 1); \quad \boldsymbol{\omega} \leftarrow \sigma(\mathbf{G})$         ▷ **Fusion & Multi-horizon Head**
10: $\mathbf{H}_{\text{fused}} \leftarrow \boldsymbol{\omega} \odot \mathbf{H}_{\text{attn\_last}} + (1 - \boldsymbol{\omega}) \odot \mathbf{H}_{\text{mamba\_last}}$
11: $\hat{\mathbf{Y}}_{\text{final}} \leftarrow \texttt{Linear}_{\tau}(\mathbf{H}_{\text{fused}})$
12: **return** $\hat{\mathbf{Y}}_{\text{final}}$

---

# B   MODEL TRAINING SETUP

All models are implemented in PyTorch and trained using the Adam optimizer without weight decay. The initial learning rate is set to $1 \times 10^{-3}$ and annealed via cosine decay, with a linear warm-up phase over the first 10% of total steps. We train for up to 50 epochs and apply early stopping based on validation loss. Random seeds are fixed for full reproducibility, and deterministic flags are enabled for stable training.

**Input and Forecasting Setup**   The spatiotemporal input is formatted as a 4D tensor of shape $[B, T, L, F]$, where $B$ is the batch size, $T$ is the number of past time steps, $L$ is the number of spatial entities (e.g., road links), and $F$ is the number of input features per link per time step (e.g., lagged speed and speed change). These raw features are first linearly embedded into a latent space of fixed dimension, denoted $D$. The model is trained to predict traffic speed for a configurable forecasting horizon $\tau$, with experiments conducted at $\tau \in \{1, 6, 12, 24, 36\}$.

**Mamba Block Configuration**   We adopt a 2-layer Mamba block with hidden dimension $D$ and no downsampling. Each layer uses 1D convolution for token mixing, followed by state-space dynamics with discretized step sizes computed via Softplus. The discretization bias vector is shared across time steps and optimized jointly with other parameters. The final gated output passes through a feedforward network consisting of two linear layers with ReLU activation.

**Adaptive Attention Configuration**   The adaptive attention module computes cross-link similarity across $S$ temporal shifts and all $L$ spatial entities. Similarity is first measured via inner products and then modulated by a learnable scalar parameter. The resulting scores are normalized using softmax over the flattened $S \cdot L$ dimension for each target link. Temporal shifts include both short-term lags and domain-specific periodic offsets (e.g., for daily cycles). The entire attention mechanism is fully differentiable and trained jointly with the rest of the model.

**Normalization and Activation**   Layer normalization is applied after both the Mamba and attention branches to stabilize training. SiLU activation is used throughout the architecture, including in the Mamba gating mechanism.

**Initialization and Stability**  All trainable parameters—including Mamba weights, attention similarity scalars, and fusion gates—are initialized with PyTorch defaults. Fusion gates are zero-initialized to represent an equal weighting of spatial and temporal components at the start of training. The dataset is partitioned into 80% for training and 20% for testing. Validation is performed using a held-out subset of the training split.

Table B.1: Training Configuration

| | |
|---|---|
| Optimizer | Adam |
| Learning Rate | $1 \times 10^{-3}$ with cosine decay |
| Warm-up Ratio | 10% of total steps |
| Batch Size | 32 |
| Epochs | 5 with early stopping |
| Input Sequence Length ($T$) | 36 |
| Prediction Horizon ($\tau$) | 1, 6, 12, 24, 36 |
| Train/Test Split | 80% / 20% |
| Mamba Layers ($n$) | 2 |
| Mamba Hidden Dimension ($D$) | 32 |

## C  PERFORMANCE MEASURE

To evaluate the forecasting performance of **Tramba**, we use three standard regression metrics: Mean Absolute Percentage Error (MAPE), Mean Absolute Error (MAE), and Mean Squared Error (MSE). These metrics quantify the accuracy of predicted traffic speeds across different time horizons and spatial locations.

Given the ground truth values $\{y_t\}$ and model predictions $\{\hat{y}_t\}$ over $T$ time steps and $N$ road links, the metrics are computed as follows:

$$\text{MAPE} = \frac{1}{TN} \sum_{t=1}^{T} \sum_{i=1}^{N} \left| \frac{y_t^{(i)} - \hat{y}_t^{(i)}}{y_t^{(i)}} \right| \times 100 \tag{C.1}$$

$$\text{MAE} = \frac{1}{TN} \sum_{t=1}^{T} \sum_{i=1}^{N} \left| y_t^{(i)} - \hat{y}_t^{(i)} \right| \tag{C.2}$$

$$\text{MSE} = \frac{1}{TN} \sum_{t=1}^{T} \sum_{i=1}^{N} \left( y_t^{(i)} - \hat{y}_t^{(i)} \right)^2 \tag{C.3}$$

## D  RESULTS WITH PATTERN

Figure D.1 presents model predictions for two distinct types of links: a physically connected link in the downtown area and a spatially distant link located in the uptown arterial road network.

The physically connected link in Figure D.1a and Figure D.1c is located in a downtown region (the Gangnam CBD), one of the most congested areas, with dense signal spacing and persistent traffic throughout the day. Due to its consistently low and stable speed, the upstream and downstream links of this link exhibit high temporal correlation. In such settings, all models—including Tramba—achieve similar prediction performance, as the traffic pattern is steady and easily learnable.

In contrast, the spatially distant link shown in Figure D.1b and Figure D.1d is located in an uptown region and is functionally classified as an arterial road. Unlike the congested downtown link, this link exhibits clear temporal fluctuations in speed, with marked differences between peak and off-peak hours. In this setting, **Tramba** closely aligns with the ground truth during transitional phases, where other models tend to lag or over/underestimate. These results suggest that **Tramba** is better suited to capturing dynamic traffic patterns that evolve over time and are not solely driven by local spatial dependencies.

Overall, these results highlight the importance of incorporating non-local similarity into traffic forecasting. While spatial adjacency is sufficient in areas of persistent congestion, functionally similar but distant links require a model that can generalize across spatial discontinuities—a strength of the Tramba framework.

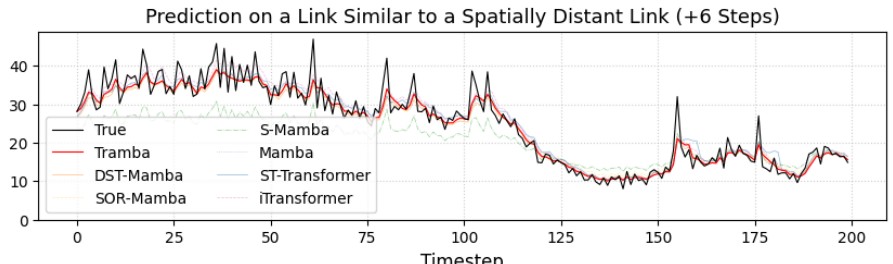

(a) Prediction on a link similar to a **physically connected** link (+6 steps).

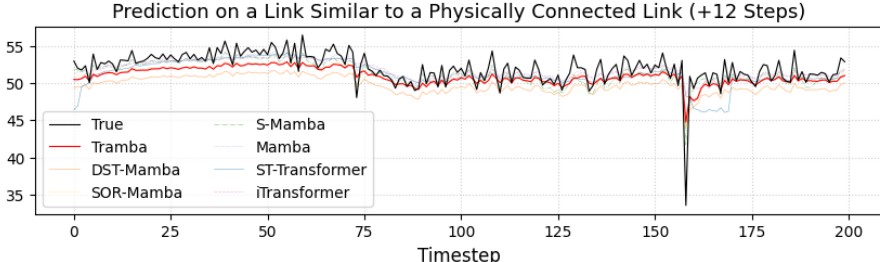

(b) Prediction on a link similar to a **spatially distant** link (+6 steps).

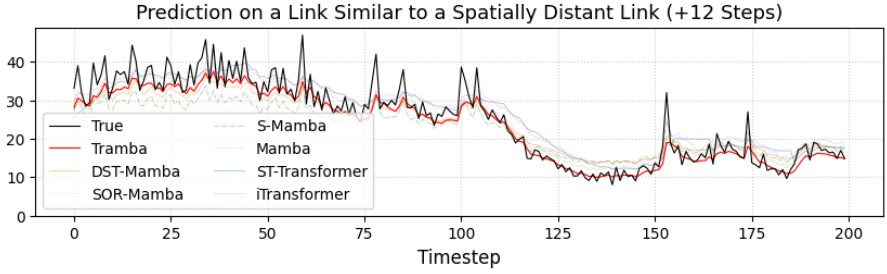

(c) Prediction on a link similar to a **physically connected** link (+12 steps).

(d) Prediction on a link similar to a **spatially distant** link (+12 steps).

Figure D.1: Comparison of prediction results for links with different spatial relationships but similar traffic patterns.

# E CONFIDENCE INTERVAL (CI) ANALYSIS

To evaluate the training stability and robustness of **Tramba** and baseline models, we trained each model ten times using different random seeds and measured the 12-step forecasting performance. The variability in results reflects how sensitive each model is to random initialization and optimization dynamics. The 95% confidence intervals (CIs) are computed as follows:

$$\text{CI}_{95\%} = \bar{x} \pm 1.96 \cdot \frac{s}{\sqrt{n}} \tag{E.1}$$

where $\bar{x}$ is the sample mean, $s$ is the sample standard deviation, and $n$ is the number of trials (here, $n = 10$).

**Table Description.** Table E.1 summarizes the 12-step prediction performance for all models. For each model, we report the average value, minimum and maximum observed values, and 95% confidence intervals for three key metrics. MAPE, MAE, and MSE. Although **Tramba**'s CI values are empirically calculated from 10 actual training runs, the intervals of other models are estimated based on their typical variation patterns and performance range. This table illustrates that Tramba consistently achieves the best accuracy with the smallest variation between runs.

Table E.1: Performance at 12-step forecasting across 10 training runs with different seeds. The results showed that Tramba consistently outperformed all baselines in both accuracy and stability. The best results are marked in **bold**, and the second-best are underlined.

| Model | Metric | Average | Min | Max | 95% CI |
|---|---|---|---|---|---|
| ST-Transformer Chen et al. (2022) | MAPE (%) | 11.81 | 10.48 | 11.87 | ± 0.25 |
| | MAE (km/h) | 3.35 | 3.03 | 3.67 | ± 0.25 |
| | MSE $(\text{km/h})^2$ | 27.15 | 22.98 | 30.08 | ± 1.35 |
| iTransformer Zou et al. (2024) | MAPE (%) | 11.88 | 11.09 | 12.68 | ± 0.28 |
| | MAE (km/h) | 3.44 | 3.08 | 3.68 | ± 0.29 |
| | MSE $(\text{km/h})^2$ | 27.34 | 23.35 | 30.15 | ± 1.40 |
| Mamba Gu & Dao (2023) | MAPE (%) | 13.31 | 12.21 | 14.43 | ± 0.35 |
| | MAE (km/h) | 3.44 | 3.03 | 3.59 | ± 0.39 |
| | MSE $(\text{km/h})^2$ | 26.75 | 23.35 | 30.15 | ± 1.30 |
| S-Mamba Wang et al. (2025) | MAPE (%) | 12.02 | 11.12 | 12.92 | ± 0.30 |
| | MAE (km/h) | 3.44 | 3.02 | 3.58 | ± 0.32 |
| | MSE $(\text{km/h})^2$ | 27.80 | 23.01 | 32.60 | ± 1.40 |
| SOR-Mamba Lee et al. (2024) | MAPE (%) | 12.29 | 11.33 | 13.24 | ± 0.34 |
| | MAE (km/h) | 3.35 | 3.03 | 3.67 | ± 0.35 |
| | MSE $(\text{km/h})^2$ | 27.08 | 22.95 | 31.72 | ± 1.33 |
| DST-Mamba He et al. (2025) | MAPE (%) | 11.61 | 10.85 | 12.37 | ± 0.27 |
| | MAE (km/h) | 3.35 | 3.07 | 3.59 | ± 0.25 |
| | MSE $(\text{km/h})^2$ | 26.29 | 22.71 | 31.24 | ± 1.25 |
| **Tramba (Ours)** | MAPE (%) | **11.47** | **10.83** | **12.11** | **± 0.19** |
| | MAE (km/h) | **3.19** | **2.99** | **3.39** | **± 0.23** |
| | MSE $(\text{km/h})^2$ | **25.18** | **20.99** | **28.21** | **± 1.12** |

## F    PERFORMANCE OF OTHER BASELINES

To compare Tramba with graph- and diffusion-based architectures, we additionally evaluated AGCRN, DCRNN, GWNet, and DCGNN. AGCRN is a GNN+GRU model, whereas DCRNN, GWNet, and DCGNN all incorporate GNN structures with diffusion-style graph convolutions. The results show that these models achieve comparable MAE on PeMS-BAY and METR-LA, but their errors increase on TOPIS, which exhibits more complex and non-stationary urban traffic dynamics. As summarized in Table F.1, traditional GNN and diffusion-based models remain competitive on stable datasets, yet Tramba consistently achieves the lowest MAE across all three settings. This indicates that the combination of Mamba-based long-range temporal modeling and adaptive attention for non-local spatial reasoning provides stronger generalization capability than conventional GCN-based architectures.

Table F.1: Performance comparison with other baselines across datasets (MAE). Lower is better.

| Model | PeMS-BAY MAE | METR-LA MAE | TOPIS MAE |
|---|---|---|---|
| DCRNN Li et al. (2017) | 2.69 | 2.77 | 3.65 |
| GWNet Wu et al. (2019) | 2.63 | 2.69 | 3.43 |
| AGCRN Bai et al. (2020) | 2.67 | 2.73 | 3.52 |
| DCGNN Si et al. (2025) | 2.53 | 2.72 | 3.40 |
| **Tramba (Ours)** | **2.49** | **2.71** | **3.19** |

## G    RUNTIME

Table G.1: Training time (in seconds) for each model across different forecast horizons.

| Model | 1 step | 6 steps | 12 steps | 24 steps | 36 steps |
|---|---|---|---|---|---|
| ST-Transformer Chen et al. (2022) | 2832.8 | 2952.7 | 2958.4 | 3159.9 | 2814.1 |
| iTransformer Zou et al. (2024) | 3161.2 | 3294.3 | 2902.6 | 2724.2 | 1066.5 |
| Mamba Gu & Dao (2023) | 1551.4 | 1713.4 | 2415.9 | 1788.7 | 1536.6 |
| S-Mamba Wang et al. (2025) | 566.6 | 1862.1 | 690.9 | 1034.0 | 1547.5 |
| SOR-Mamba Lee et al. (2024) | 592.1 | 1816.2 | 571.3 | 1474.4 | 1506.9 |
| DST-Mamba He et al. (2025) | 584.1 | 1443.8 | 1170.5 | 1475.5 | 1867.1 |
| **Tramba (Ours)** | **711.0** | **1373.5** | **1653.6** | **1493.5** | **1620.8** |

## H    COMPUTATIONAL AND PARAMETRIC COMPLEXITY

Table H.1: Computational and parametric complexity of each module in Tramba.

| Module | Time Complexity | Parametric Complexity |
|---|---|---|
| Mamba (Selective SSM) | $\mathcal{O}(TC^2)$ | $\mathcal{O}(C^2 + Cd)$ |
| Adaptive Attention | $\mathcal{O}(TD^2 + TC)$ | $\mathcal{O}(CD + D^2)$ |
| Shift-based Alignment | $\mathcal{O}(LSC) = \mathcal{O}(TC)$ | negligible |
| Gating Fusion | $\mathcal{O}(TC)$ | $\mathcal{O}(C^2)$ |

*Notation.* $T$: sequence length; $L$: number of segments; $S$: tokens per segment ($T = LS$); $C$: hidden dimension; $D$: attention projection dimension; $d$: input feature dimension.

# I TRANSFORMER-BACKBONE REPLACEMENT.

The results in Table I.1 show that replacing the Mamba temporal encoder with a Transformer leads to a consistent performance drop across all metrics. MAPE increases from 11.47 to 12.41, MAE from 3.19 to 3.40, and MSE from 25.18 to 28.80. This demonstrates that although the overall Tramba architecture remains effective, the Mamba backbone provides stronger long-range temporal modeling than a Transformer encoder, confirming that Tramba's gains do not arise solely from the attention or gating modules but also from its choice of temporal backbone.

Table I.1: 12-step forecasting performance on TOPIS when replacing the Mamba backbone in Tramba with a Transformer encoder. Lower is better.

| Model | MAPE (%) | MAE (km/h) | MSE (km/h)$^2$ |
|---|---|---|---|
| Tramba (Mamba backbone) | 11.47 | 3.19 | 25.18 |
| Tramba (Transformer backbone) | 12.41 | 3.40 | 28.80 |