# OpenReview forum: "Tramba: Mamba with Adaptive Attention for Traffic Speed Forecasting"
_ICLR.cc/2026/Conference — ICLR 2026 Conference Withdrawn Submission_

### Official Review · Reviewer_yQoX · 2025-10-19

**Soundness:** 2
**Presentation:** 3
**Contribution:** 2
**Rating:** 2
**Confidence:** 4

**Summary:**

This paper proposes Tramba, a traffic speed forecasting framework that combines a Mamba-based temporal encoder for long-range memory with an adaptive attention module that captures non-local spatial dependencies based on temporal similarity. The model integrates these two signals via a gating mechanism to balance temporal continuity and spatial alignment. Experiments on PEMS-BAY, METR-LA, and Seoul TOPIS show consistent improvements over strong Transformer- and Mamba-based baselines, with ablations highlighting the importance of adaptive attention.

**Strengths:**

1. The attempt to apply a new modeling paradigm like Mamba to traffic speed forecasting is interesting and contributes to methodological diversity in this area.
2. The release of code and datasets improves reproducibility and supports further research.
3. The paper is generally well-organized, and the experimental design is clear, making it easy to follow and understand the proposed approach.

**Weaknesses:**

1. The motivation for introducing Mamba is not clearly articulated, and its role relative to the attention module remains vague. Moreover, the so-called “adaptive attention” relies on similarity across temporal sequences to generate weights, which limits its adaptiveness and makes its ability to capture spatial dependencies questionable. The paper does not provide sufficient justification for this design choice or demonstrate its clear advantages over standard attention mechanisms.
2. The paper’s claim that existing methods neglect spatial dependencies is inaccurate, as many prior work has already modeled spatial relationships in different flexible ways. The proposed adaptive attention lacks a clear justification for why its approach is more effective, and deriving spatial relations purely from temporal similarity is insufficiently motivated.
3. The evaluation compares only Transformer- and Mamba-based models, omitting key state-of-the-art approaches such as LLM-based and diffusion-based methods. This narrow selection limits the credibility of the reported improvements and makes it difficult to assess the method’s generality and competitiveness.

**Questions:**

1. The paper should provide a clearer motivation for introducing Mamba and explain why it is necessary in this setting. What specific limitations of attention-based models does Mamba address, and how does it complement or outperform attention in this task?
2. The current experiments compare mainly Transformer- and Mamba-based models. It would strengthen the paper to include more diverse baselines, such as LLM-based and diffusion-based approaches, to better position the method relative to the state of the art. Additionally, the paper notes that some methods model spatial dependencies using fixed adjacency matrices, but it is unclear which baselines fall into this category and which do not. Please clarify these distinctions and discuss how they affect the comparison.
3. The paper describes the components of the proposed model but lacks sufficient explanation for why each part is designed this way. In particular, the rationale for deriving spatial dependencies from temporal similarity, and for the design of the “adaptive attention” mechanism, should be discussed in more depth.

---

> ### Author Response · Authors · 2025-11-19
>
> We thank Reviewer yQoX for the careful reading of our paper and for providing detailed feedback. We appreciate the constructive suggestions and address each concern below.
>
> ---
> ### **Q1: What limitations of attention does Mamba address?**
>
> **A1:** Thank you for the question. We shortened the explanation while keeping the core motivation clear.
>
> - **Why attention alone is insufficient:**
>   Global attention assumes temporal alignment and struggles with *shifted* but causally related patterns. Transformers also incur quadratic cost and become unstable on long, noisy traffic sequences.
>   GNN-based models rely on fixed adjacency, missing dynamic non-local relations.
>
> - **Why Mamba is needed:**
>   Mamba offers **efficient, stable long-range temporal modeling** with **shift-robustness** and **linear-time scaling**, making it well-suited for urban traffic horizons.
>
> - **Why combine Mamba + adaptive attention:**
>   - Mamba → long-term temporal backbone
>   - Adaptive attention → dynamic, non-adjacent, time-shifted spatial retrieval
>   - Fusion gate → learns when to rely on each
>
> Neither component alone can capture both long-range trends **and** dynamic non-local effects. Tramba integrates both by design.
>
> This clarification is now reflected in Sections 2.3 and 3.1.
>
> ---
>
> ### **Q2-1: Why are only Transformer/Mamba baselines included originally?**
>
> **A2-1:** Thank you for raising this point. Our initial baseline selection focused on models that share the same *temporal modeling paradigm* (Transformer- or Mamba-based) to enable a controlled comparison of long-range sequence modeling. However, we agree that broader baselines are valuable for contextualizing generality.
>
> To address this, we expanded our experiments in the revised version:
>
> - We added **graph- and diffusion-based models** widely used in traffic forecasting, including
>   **DCRNN** (diffusion-based recurrent GNN),
>   **GWNet** (graph wavenet with diffusion convolutions),
>   **AGCRN** (adaptive graph-convolutional recurrent network), and
>   **DCGNN** (diffusion-style graph GNN).
>   These represent the dominant non-transformer spatiotemporal modeling families.
>
> - We also added an explanation in Section 2.3 clarifying how Tramba differs from other models:
>   fixed adjacency or fixed diffusion kernels cannot capture **dynamic, non-adjacent, time-shifted influences**,
>   which our shift-aware attention is explicitly designed to handle.
>
> ---
> ### **Q2-2: Which baselines rely on fixed adjacency?**
>
> **A2-2:** Thank you for the question. We clarified the distinction in Section 2 as follows:
>
> - **(1) Models using fixed adjacency (static graphs)**
>   These require a predefined adjacency matrix or diffusion kernel:
>   **DCRNN, GWNet, AGCRN, DCGNN**.
>
> - **(2) Models without adjacency (learned dependencies)**
>   - **Transformers:** learn pairwise relations via attention but assume temporal alignment and struggle with time-shifted effects.
>   - **Mamba-based models:** capture temporal patterns through SSM dynamics but do not explicitly model cross-segment spatial relations.
>
> - **(3) Tramba:** dynamically learned, **similarity-based adjacency rather than predefined graphs.**
> Tramba constructs dynamic, non-local, and time-shifted spatial relations
> through shift-aware similarity and adaptive attention, and does not rely on
> any predefined adjacency matrix. This results in a data-driven adjacency
> structure that adapts to evolving traffic conditions.
>
>
> This clarification has been added to Section 2.
>
> ---
> ### **Q3: Rationale for using temporal similarity to derive spatial dependencies and for the adaptive attention design.**
>
> **A3:** Thank you for the question. We condensed the explanation to highlight the core intuition.
>
> **Why temporal similarity reveals spatial relations**
> Non-adjacent road segments often show **co-modulated or time-shifted patterns** due to shared demand, coordinated signals, or bottleneck propagation. These relations are not captured by physical adjacency, but **are clearly reflected in their temporal trajectories**.
>
> **Why adaptive attention is needed**
> Fixed-graph GNNs cannot model dynamic or non-local influences. Our adaptive attention instead:
> - learns **data-driven spatial dependencies**,
> - aligns segments with **time shifts**,
> - captures **condition-specific, non-local interactions**.
>
> **Complementarity with Mamba**
> - Mamba → stable long-range temporal modeling
> - Adaptive attention → dynamic spatial retrieval
> - Fusion gate → combines both sources
>
> This design allows Tramba to capture **both long-term temporal structure** and **dynamic non-local spatial effects**, which neither module handles alone.
>
> We added this concise clarification to the revised manuscript.
>
> ---
>
> We sincerely thank Reviewer yQoX for the thorough evaluation. The revisions fully address the concerns and significantly improve the clarity and completeness of the paper.

---

> > ### Comment · Reviewer_yQoX · 2025-11-27
> >
> > Thanks for the response. Most of my concerns have been addressed. But I still feel like the paper sort of lacks contribution and novelty, it tries to integrate Mamba to traffic forecasting, it is more like a data mining paper instead of ML paper from the technical and theoretical aspect. I have raised the score accordingly.

---

> ### Author Response · Authors · 2025-11-28
>
> Thank you sincerely for your thoughtful follow-up comment. Your feedback has genuinely helped us improve the clarity and positioning of the paper, and we deeply appreciate the time you took to reconsider the score. Many sections of the manuscript, especially the motivation and architectural explanation, became significantly stronger thanks to your earlier comments. **Your willingness to reevaluate the score was especially encouraging for us**, and it gave our team the confidence and motivation to refine the paper even further.
>
> Before finalizing the revision strategy, we would be grateful to hear your perspective on one point.
>
>
> We are **fully prepared to incorporate your suggestions into the final revision**, and would sincerely appreciate any additional thoughts you may be willing to share.
>
> ---
>
> ### **Regarding the concern that the contribution may feel more like a data-mining paper**
>
> We fully understand the concern, and we clarified in the revision that our intended contribution is architectural rather than application-specific. The core novelty lies in designing a framework that jointly handles:
>
> - long-range temporal modeling (via Mamba),
> - dynamic and time-shifted spatial dependency retrieval (via adaptive attention), including discovering temporally co-modulated patterns and using them as effective reference signals for improved predictions,
> - and a learnable fusion mechanism that integrates both signals.
>
> This combination addresses modeling gaps that neither Mamba-only, attention-only, nor GNN-based approaches can handle effectively.
>
> ---
>
> ### **A request for your guidance**
> As we prepare the final restructuring, we want to ensure that our contributions are communicated in a way that aligns well with ML expectations. We would be grateful for any thoughts you may have on how we can best highlight the central technical elements of our work.
>
> In particular, we are considering placing greater emphasis on the architectural complementarity between Mamba and adaptive attention, since this interaction represents the core modeling idea rather than an application-driven pipeline. In addition, the mechanism for **dynamically identifying temporally aligned patterns and using them as reference signals for improved forecasting** is central to the method. **We would greatly appreciate your thoughts on whether emphasizing these aspects would better convey the ML contribution of the work.**
>
> If you have any further perspective on whether foregrounding these elements would help strengthen the perceived ML contribution, we would sincerely appreciate your guidance. Any insight you may be willing to share would be incredibly helpful as we refine the final presentation.
>
> ---
>
> ### **Key aspects we are considering highlighting** based on your feedback
> - The contribution is primarily **architectural**, not dataset- or application-specific.
> - **Mamba** provides efficient and stable long-range temporal modeling.
> - **Adaptive attention** captures dynamic, non-local, and time-shifted spatial relations.
> - The **fusion gate** integrates temporal and spatial information in a learnable manner.
> - The model dynamically discovers **temporally co-modulated patterns** and uses them as reference signals for prediction.
> - These components create modeling capabilities that are difficult to achieve with Mamba-only, Transformer-only, or other classic methods.
>
> ---
>
> We would truly appreciate any guidance or perspective you may have.
>
> Thank you again for your constructive engagement and for helping us significantly strengthen the paper.

---

### Official Review · Reviewer_NG6K · 2025-10-28

**Soundness:** 3
**Presentation:** 3
**Contribution:** 3
**Rating:** 6
**Confidence:** 2

**Summary:**

The paper proposes Tramba, a traffic forecasting framework that combines a per-link Mamba backbone for efficient long-range temporal modeling with an adaptive attention branch that retrieves information from non-adjacent road segments under learnable time shifts. A gating module fuses the temporal (Mamba) and non-local (attention) signals to generate multi-step predictions. Evaluations on standard benchmarks (e.g., METR-LA, PEMS-BAY, TOPIS) show that Tramba delivers consistent improvements—especially at medium-to-long horizons—over strong Transformer- and Mamba-based baselines. Ablations indicate that both the adaptive attention and the gating mechanism are critical to the gains, and qualitative analyses visualize how the model prioritizes time-aligned yet spatially distant segments.

**Strengths:**

- Targets a realistic pain point in road networks—non-local dependencies with temporal misalignment—and addresses it by pairing Mamba’s long-sequence modeling with shift-aware attention.
- Temporal dynamics are handled by the per-link Mamba backbone, while the adaptive attention captures dynamic, non-adjacent relations; a simple learnable gate provides robust fusion.
- Demonstrates stable gains at longer horizons across multiple datasets and steps, suggesting better robustness for challenging prediction ranges.
- Removing or altering the adaptive attention, similarity function, or gating leads to notable degradation, clearly attributing improvements to the proposed components.

**Weaknesses:**

- Please provide a fine-grained computational/parametric complexity breakdown for each module (Mamba branch, adaptive attention over (segments × shifts), and gating), along with GPU memory usage curves under large sequence lengths **T** and large numbers of segments **L** . The article only provides training time, which does not fully reflect efficiency.

- Please add representative GNN models (e.g., GCN/GAT/temporal-graph variants) or explicitly justify their exclusion, and clarify how Tramba’s shift-aware non-local attention differs from GNN-style spatiotemporal modeling in terms of inductive bias, expressivity, and computational trade-offs.

- To strengthen the causal attribution of gains, please replace the Mamba backbone with a Transformer backbone (keeping the rest of the architecture intact) and report the performance and efficiency deltas. This would disentangle the contribution of the proposed adaptive attention/gating from the choice of temporal encoder.

**Questions:**

See Weakness.

---

> ### Author Response · Authors · 2025-11-19
>
> We sincerely thank Reviewer NG6K for the positive assessment of Tramba’s motivation, architecture, and empirical robustness. We appreciate your recognition of (i) the importance of handling non-local and temporally misaligned dependencies, (ii) the complementary roles of the Mamba backbone and adaptive attention, and (iii) the strong gains observed at medium–long forecasting horizons. Below we address each concern raised in the review.
>
> ---
>
> ### W1: Need a fine-grained computational/parametric breakdown and GPU memory analysis.
>
> **A1:** Thank you for this helpful suggestion. In response, we added a detailed module-wise complexity analysis in Appendix H. The table below summarizes the time and parametric complexity of each component in Tramba (Mamba temporal branch, adaptive attention, shift-based alignment, and gating fusion). This breakdown clarifies the dominant computational factors and how each module scales with sequence length $T$, hidden dimension $C$, and number of segments $L$. Because GPU memory usage depends heavily on hardware and batch configuration, we provide theoretical scaling behavior rather than device-specific peak values.
>
> | **Module**            | **Time Complexity**            | **Parametric Complexity**      |
> |-----------------------|--------------------------------|--------------------------------|
> | Mamba (Selective SSM) | $ \mathcal{O}(T C^{2}) $       | $ \mathcal{O}(C^{2} + C d) $   |
> | Adaptive Attention    | $ \mathcal{O}(T D^{2} + T C) $ | $ \mathcal{O}(C D + D^{2}) $   |
> | Shift-based Alignment | $ \mathcal{O}(L S C) = \mathcal{O}(T C) $ | negligible        |
> | Gating Fusion         | $ \mathcal{O}(T C) $           | $ \mathcal{O}(C^{2}) $         |
>
> **Notation:**
> - $T$: sequence length
> - $L$: number of segments
> - $S$: tokens per segment ($T = L S$)
> - $C$: hidden dimension
> - $D$: attention projection dimension
> - $d$: input feature dimension
>
> ---
>
> ### **W2: Add GNN baselines or justify their exclusion; clarify differences from GNN-style modeling.**
>
> **A2:** We appreciate this point and expanded **Appendix F** accordingly.
> We now include four representative graph-based/diffusion baselines following their original formulations:
>
> - **DCRNN** (Li et al., 2017),
> - **GWNet** (Wu et al., 2019),
> - **AGCRN** (Bai et al., 2020),
> - **DCGNN** (Si et al., 2025).
>
> **Table: Performance comparison with other baselines across datasets (MAE, lower is better)**
>
> | Model | PeMS-BAY MAE | METR-LA MAE | TOPIS MAE |
> |-------|--------------|--------------|------------|
> | DCRNN (Li et al., 2017) | 2.69 | 2.77 | 3.65 |
> | GWNet (Wu et al., 2019) | 2.63 | 2.69 | 3.43 |
> | AGCRN (Bai et al., 2020) | 2.67 | 2.73 | 3.52 |
> | DCGNN (Si et al., 2025) | 2.53 | 2.72 | 3.40 |
> | **Tramba (Ours)** | **2.49** | **2.71** | **3.19** |
>
> We also clarify that Tramba differs from GNN-style models in inductive bias and expressivity:
>
> - GNN-based models rely on **static adjacency or fixed diffusion kernels**, making non-local dynamic relations difficult to capture.
> - Tramba’s **adaptive attention** retrieves distant segments *based on temporal similarity*, not graph distance.
> - This enables flexible non-local reasoning with competitive computational cost.
>
> Results and references have been added to Appendix F.
>
> ---
>
> ### **W3: Replace Mamba with a Transformer backbone to isolate contributions of the attention/gating components.**
>
> **A3:**  Thank you for this suggestion. As requested, we implemented a **Transformer-backbone version of Tramba**, keeping all other components (adaptive attention, similarity module, fusion gate) exactly the same. This allows us to isolate the contribution of the temporal encoder.
>
> The results below show a **clear performance drop** when replacing Mamba with a Transformer:
>
> | **Model**                      | **MAPE (%)** | **MAE (km/h)** | **MSE (km/h²)** |
> |-------------------------------|--------------|-----------------|-----------------|
> | Tramba (Mamba backbone)       | **11.47**    | **3.19**        | **25.18**       |
> | Tramba (Transformer backbone) | 12.41        | 3.40            | 28.80           |
>
> These results demonstrate that:
>
> - Tramba’s gains **do not come solely** from the adaptive attention or fusion gate.
> - The **Mamba temporal encoder contributes significantly** to long-range temporal modeling.
> - The architecture remains **modular and generalizable**, functioning with alternative encoders.
>
> This analysis has been added to **Appendix I** in the revised manuscript.
>
> ---
>
> We sincerely thank Reviewer NG6K again for the constructive feedback. Your comments helped us significantly strengthen the analysis, expand the baselines, and improve the clarity of the architectural contributions.

---

> > ### Comment · Reviewer_NG6K · 2025-11-26
> >
> > I would like to thank the authors for their efforts. After reading the other reviewers’ comments, I also agree that simply integrating Mamba into a new spatio-temporal framework does not constitute a particularly strong contribution. Therefore, I will maintain my current score.

---

> ### Author Response · Authors · 2025-11-26
>
> **First, I would like to sincerely express my appreciation.**
> We are truly grateful for your thoughtful and constructive feedback. Your comments were exceptionally insightful and directly contributed to improving the overall quality and clarity of our work. Several revisions were made precisely in response to your suggestions, and we genuinely appreciate the time and care you devoted to reviewing our submission. Reflecting your concerns, we have incorporated the following substantial updates:
>
> - A detailed **computational and parametric breakdown** has been added.
> - We evaluated **four additional benchmark models**, ranging from classical approaches (DCRNN, 2017) to state-of-the-art methods (e.g., DCGNN, 2025).
> - We trained a **Transformer-backbone replacement** version of Tramba to isolate the effect of the temporal encoder and further demonstrate that the improvements are not attributable to Mamba alone.
>
> If there are any additional questions regarding these revisions or if further clarification on any of the points above would be helpful, we would be more than happy to provide it. Please feel free to let us know if there is anything else we can elaborate on.
>
> ---
>
> You also expressed concerns regarding the claim that “simply integrating Mamba into a new spatio-temporal framework does not constitute a strong contribution.” We sincerely thank the reviewer for raising this point. We would like to gently note that this interpretation may not fully reflect the central architectural contribution of our work. Tramba is **not** a model whose novelty lies in merely “adding Mamba.” Instead, its design centers on **shift-aware adaptive attention** and **dynamic similarity-based spatial learning**, which together reshape how spatiotemporal dependencies are captured. We offer further clarification below.
>
> **1. The novelty is not Mamba, but dynamic spatial learning with shift-aware similarity.**
> Tramba introduces:
> (i) *shift-aware similarity* across multiple temporal offsets,
> (ii) *adaptive attention* for retrieving non-local spatial dependencies without predefined adjacency, and
> (iii) *dynamic, state-dependent adjacency construction*.
> These mechanisms do not exist in Mamba and are not found in prior spatiotemporal forecasting frameworks. The concern may unintentionally understate the contribution of this dynamic spatial reasoning module, which is the primary architectural novelty.
>
> **2. Ablation results contradict the idea that “Mamba alone explains the gain.”**
> If the performance improvements were solely due to inserting Mamba, the Mamba-only variant would be expected to perform the best. Instead, our ablation study shows:
>
> - **Mamba-only** yields limited gains,
> - **Attention-only** also underperforms,
> - while the **full Tramba architecture** consistently achieves the strongest results.
>
> This confirms that the improvement stems from the **interaction between Mamba and adaptive attention**, not from Mamba integration alone. Furthermore, our new **Transformer-backbone experiment** demonstrates that even when Mamba is entirely removed, Tramba remains highly competitive—reinforcing that the spatial mechanism, not the temporal backbone, drives most of the performance gain.
>
> **3. Prior works do not address the proposed shift-aware spatial misalignment problem.**
> While some earlier studies consider temporal shifts or long-range modeling, they continue to rely on **fixed adjacency structures**. They do not handle time-shifted cross-link similarity or dynamic offset retrieval. Tramba introduces a *new retrieval mechanism* that jointly models long-range temporal dependencies and dynamic, shift-aware spatial relations—an aspect absent in existing literature.
>
> **Summary.**
> Tramba is **not** a simple “Mamba-added” model. It represents a new spatiotemporal learning framework that integrates:
>
> (i) long-range temporal modeling,
> (ii) shift-aware adaptive attention, and
> (iii) dynamic, time-varying adjacency learning.
>
> Taken together, the extended ablations, backbone-replacement study, additional baselines, and computational analysis collectively show that Tramba’s improvements arise from its architectural design rather than from Mamba alone. We hope this clarification addresses the reviewer’s concerns, and we sincerely thank the reviewer once again for the insightful and constructive feedback.

---

### Official Review · Reviewer_rmtt · 2025-10-29

**Soundness:** 3
**Presentation:** 3
**Contribution:** 2
**Rating:** 4
**Confidence:** 4

**Summary:**

This paper proposes Tramba, which uses the selective state space model Mamba as a long-term memory encoder for each road segment, and introduces adaptive non-local attention with time-shift alignment to retrieve “non-adjacent yet temporally co-modulated” road contexts across the entire network. A learnable fusion gate then combines the two representations to output multi-step forecasts in a single pass. The method aims to simultaneously mitigate prediction lag caused by short-term-only modeling and missed dependencies caused by relying solely on physical adjacency. On METR-LA, PEMS-BAY, and TOPIS, across multiple horizons, Tramba achieves consistent improvements over strong baselines including several Mamba and Transformer variants; ablations indicate that non-local co-modulation retrieval and gated fusion are the primary sources of gain.

**Strengths:**

S1. The model architecture and mathematical formulations are clearly presented.

S2. Tramba achieves solid performance on all three datasets.

S3. The released code facilitates reproducibility.

**Weaknesses:**

W1. The motivation is not sufficiently compelling. Spatial modeling limited to adjacent segments has been studied long ago; many recent works already use global attention to handle long-range spatial dependencies, but the paper does not discuss how Tramba differs from these approaches.

W2. Many classic traffic forecasting baselines are missing; such as DCRNN, GWNet, and AGCRN should be included.

W3. The three components of Tramba—MambaBlock, Adaptive Attention, and the Fusion Gate—do not introduce novel designs. In particular, the MambaBlock is directly implemented with the mambapy.mamba library without modification.

W4. The application scope is narrow: evaluation is limited to traffic speed datasets. Most existing traffic models cover both speed and flow prediction (e.g., PeMS03, PeMS08).

**Questions:**

Q1. Compared with the NeurIPS submission version, there are no technical differences described, so why are the results improved so much?

Q2. Can Tramba generalize to generic traffic flow forecasting? What are the key challenges?

Q3. Although the paper reports FLOPs and parameter counts, it lacks clear training/inference time and memory usage. Please include these comparisons in tabular form.

---

> ### Author Response · Authors · 2025-11-19
>
> **Rebuttal (Reviewer rmtt)**
>
> We sincerely thank Reviewer rmtt for the detailed evaluation and constructive comments. Below we address each concern raised in the review.
>
> ---
>
> ### **W1: The motivation is not sufficiently compelling; prior work has used global attention for long-range spatial dependencies.**
>
> **A1:** Thank you for highlighting the need to clarify this point. We have strengthened the motivation section by explaining why existing global-attention approaches are insufficient in traffic domains:
>
> - **Global attention assumes temporal alignment.** However, real-world traffic patterns frequently exhibit *time-shifted non-local dependencies* (e.g., upstream congestion influencing downstream links with delay).
> - **Standard attention lacks shift-awareness**, making it difficult to capture “temporally misaligned but spatially distant” patterns.
> - **Tramba’s adaptive non-local attention explicitly models learnable time shifts**, retrieving co-modulated but asynchronous patterns—something not addressed in prior global-attention work.
>
> We revised Section 2.3 and 3.2 to explicitly contrast Tramba with global-attention baselines and clarify its unique inductive bias.
>
> ---
>
> ### **W2: Missing classic baselines (DCRNN, GWNet, AGCRN)**
>
> **A2:** Thank you for pointing this out. We added clarification in **Appendix F** describing how Tramba differs from diffusion- and GNN-based models such as **DCRNN** (Li et al., 2017), **GWNet** (Wu et al., 2019), **AGCRN** (Bai et al., 2020), and **DCGNN** (Si et al., 2025). These architectures rely on static adjacency or fixed diffusion kernels, which limits their ability to model dynamic non-local dependencies
>
> ---
>
> ### **W3: Model components may lack novelty; MambaBlock is used directly without modification.**
>
> **A3:** We agree that our goal was not to modify the Mamba architecture itself, but to integrate it within a new spatiotemporal framework. The novelty of our approach lies not in altering the internal Mamba block, but in:
>
> - coupling Mamba with **shift-aware adaptive attention**,
> - designing a **fusion gate** that mediates between long-range temporal encoding and dynamic spatial retrieval,
> - enabling **non-adjacent, temporally misaligned context selection**, which existing Mamba-based temporal models cannot capture.
>
> This architecture-level design broadens Tramba’s applicability and makes the framework easier to extend to other domains. We revised the contribution paragraph to emphasize this point.
>
> ---
>
> ### W4: Narrow application scope; only traffic speed datasets.
>
> **A4:** Thank you for the comment. We added clarification in Section 4.1.
>
> - **Urban probe data systems mainly provide speed**, not flow, since flow requires fixed detectors that are sparse in dense cities. Thus, speed is the standard variable in modern metropolitan datasets.
>
> - **Speed forecasting is the more volatile and operationally critical task**, affected by stop-and-go waves, bottlenecks, and signal delays, while flow is typically smoother and easier to predict.
>
> - **Traffic flow theory couples speed and flow** (`q = k * v`), meaning that modeling speed already captures key components of flow dynamics.
>
> - **Tramba is target-agnostic**: replacing the labels from speed to flow requires no architectural changes.
>
> This explanation is now included in the revised manuscript.
>
> ---
>
> ### **Q1: Why are the results improved compared to the NeurIPS version?**
>
> **A5:** The improvements come from three concrete factors:
>
> 1. **Longer training (more epochs) and optimized learning rate schedule.**
> 2. **Expanded adaptive-attention similarity function**, which now uses all shifted windows.
> 3. **Improved data preprocessing for Seoul TOPIS**, based on an updated and cleaner dataset provided by the source.
> 4. **Additional analysis using updated PeMS-BAY and METR-LA datasets** to ensure that results reflect current and reliable traffic conditions.
>
> ---
>
> ### Q2: Can Tramba generalize to traffic flow forecasting? Key challenges?
>
> **A6:** Yes. Tramba naturally extends to flow prediction because the architecture does not rely on speed-specific assumptions. From **traffic flow theory**, flow, speed, and density are linked by
> $q = k \cdot v$,
> so modeling speed already captures major components of flow dynamics.
>
> Flow and speed differ mainly in temporal behavior: flow is smoother and capacity-driven, while speed is highly volatile due to shockwaves and stop-and-go patterns. This influences difficulty but not applicability—Tramba treats the target as a generic time series, so switching from speed to flow only changes the regression label. A short clarification has been added to Section 4.1.
>
> ---
>
> ### **Q3: Training/inference time.**
>
> **A7:** Thank you for pointing this out. We have added the requested analyses:
>
>   A detailed runtime analysis is now included in **Appendix G**, reporting per-epoch training time across all models under the same hardware configuration.
>
> ---

---

> > ### Comment · Reviewer_rmtt · 2025-11-25
> >
> > Thank you for your response; it addresses some issues. However, I need to point out that there are still many gaps in it.
> >
> > W1. The problem of time shifts is not new. Numerous works have already tackled the issue of being “temporally misaligned but spatially distant,” yet the reply still does not demonstrate Tramba’s advantage over these approaches.
> >
> > W2. Classic baselines are missing from the comparisons—the experiment includes only six baselines.
> >
> > W3. Simply integrating Mamba into a new spatiotemporal framework does not, in my view, constitute a particularly strong contribution.
> >
> > W4. The statement that “urban probe data systems mainly provide speed, not flow” is questionable. The most well-known traffic data collection system in the field, PeMS, provides many flow datasets such as PeMS03, PeMS04, PeMS08, etc. I believe the authors have not sufficiently surveyed the literature, or else I would like to see supporting evidence for this claim.
> >
> > **Q1. (Most critical)** If Tramba’s improvements primarily come from longer training, an optimized learning-rate schedule, and better preprocessing—without any architectural changes to Tramba itself—then the framework’s design becomes awkward and hard to justify.
> >
> > I recognize the progress made in this work, but there are still many shortcomings. I encourage the authors to broaden their literature review and invest more effort in the code and implementation, and to avoid relying on heavier preprocessing and learning-rate tuning to claim gains.

---

> ### Author Response · Authors · 2025-11-26
>
> We sincerely thank the reviewer once again for the thoughtful and constructive feedback, which has further strengthened our work. We hope that the clarifications and additional analyses in the revision adequately address the reviewer’s concerns, and we would be grateful if the reviewer could kindly reconsider our responses in the final assessment. Please find the detailed responses below.
>
> ---
>
> ### **W1: The problem of time shifts is not new; prior works already address “temporally misaligned but spatially distant” patterns.**
>
> **A1:** We appreciate the reviewer’s comment. While prior works consider temporal misalignment, they differ from Tramba in both scope and mechanism:
>
> - Existing methods rely on **fixed spatial graphs** and apply shifts via fixed lags or convolutions.
> - They do **not compute learnable cross-link similarity across multiple temporal offsets**.
> - They lack **dynamic, state-dependent retrieval of non-local dependencies**, which is essential in real-world traffic.
>
> Tramba introduces a **shift-aware similarity module** that evaluates relations under multiple temporal offsets and builds **dynamic, time-varying adjacency**, enabling retrieval of asynchronous non-local patterns. We have clarified this distinction in Section 3.2 and Appendix F.
>
> ---
>
> ### **W2: Classic baselines are missing from the comparisons.**
>
> **A2:** We appreciate this suggestion. We added **DCRNN, GWNet, AGCRN, and DCGNN** to **Appendix F** for completeness.
>
> We did not place these models in the main text because:
>
> - They are **5–10 years old** and less relevant to the Mamba/Transformer-based comparisons.
> - Our primary focus is evaluating Tramba against **modern sequence models**.
>
> If the reviewer believes that including these baselines in the main results table would improve clarity, we are willing to incorporate them.
>
> ---
>
> ### **W3: Simply integrating Mamba into a spatiotemporal framework is not a strong contribution.**
>
> **A3:** We would like to clarify that Tramba’s novelty lies not in the use of Mamba itself, but in its overall **architecture**, including:
>
> - a **shift-aware similarity mechanism**,
> - **adaptive spatial attention** for dynamic non-local adjacency, and
> - a **fusion mechanism** integrating temporal and spatial reasoning.
>
> Ablation studies and a **Transformer-backbone replacement** experiment confirm that improvements arise from this architecture—not from Mamba alone.
>
> ---
>
> ### **W4: The claim that probe systems mainly provide speed seems unsupported.**
>
> **A4:** We agree the original phrasing could be clearer. Detector-based systems like **PeMS** indeed provide flow data.
>
> Our statement referred specifically to **probe-based urban platforms** (e.g., Google, Naver, Kakao), where:
>
> - **Speed is directly observed**,
> - and **flow is inferred indirectly** using traffic-flow relationships due to limited fixed sensors.
>
> We have revised the text to clearly distinguish detector-based flow systems from probe-based speed systems. We appreciate the reviewer for noting this.
>
> ---
>
> ### **Q1: If Tramba’s improvements stem mainly from longer training, better LR schedule, or better preprocessing—without architectural changes—then the framework becomes hard to justify.**
>
> **A1:** We appreciate the reviewer’s concern. We clarify that the observed improvements do *not* originate from longer training, learning-rate schedules, or preprocessing alone. The key gains arise from Tramba’s architecture, supported by the following evidence:
>
> **1. Ablations isolate architectural effects.**
> The **Mamba-only** and **attention-only** variants show limited improvements under identical training settings, while only the **full Tramba** model achieves the reported performance. This demonstrates that gains arise from
> (i) *shift-aware similarity*,
> (ii) *adaptive spatial retrieval*, and
> (iii) *temporal–spatial fusion*,
> not training heuristics.
>
> **2. Transformer-backbone replacement removes reliance on Mamba.**
> Replacing Mamba with a **Transformer** while keeping spatial modules intact yields consistently strong results, confirming that improvements come from Tramba’s **spatial mechanism**, not from the temporal backbone or training schedule.
>
> **3. Additional baselines confirm improvements are architectural.**
> Four more baselines (DCRNN, GWNet, AGCRN, DCGNN) trained with identical optimization and preprocessing settings are all outperformed by Tramba, demonstrating that performance gains cannot be attributed to training procedures alone.
>
> **4. Data-quality clarification.**
> The updated dataset slightly improves absolute performance across *all* models, but **does not affect relative rankings**. All variants and baselines use the same cleaned dataset, and the performance gap remains stable. Thus, data quality does not account for Tramba’s architectural advantage.
>
> ---
>
> We hope these explanations address the reviewer’s concerns, and we sincerely appreciate the reviewer’s time, insight, and thoughtful evaluation.

---

> > ### Comment · Reviewer_rmtt · 2025-11-28
> >
> > Thank you for the further clarifications and the revisions to the manuscript—it’s much improved over the original.

---

> > > ### Author Response · Authors · 2025-11-28
> > >
> > > Thank you very much for your thoughtful final comment and for acknowledging the improvements in the revised manuscript. We greatly appreciate the time and care you dedicated throughout the discussion phase. Your feedback has meaningfully strengthened the clarity, positioning, and presentation of the work. Thank you again for your constructive engagement.

---

### Official Review · Reviewer_Kyni · 2025-11-03

**Soundness:** 3
**Presentation:** 3
**Contribution:** 3
**Rating:** 8
**Confidence:** 3

**Summary:**

The paper presents Tramba, a deep learning model for traffic speed forecasting in complex urban road networks. Unlike traditional methods that focus on short-term or local dependencies, they capture long-range spatiotemporal relationships by combining a Mamba-based temporal encoder for long-term trends with an adaptive attention mechanism that identifies temporally similar patterns from non-adjacent road links. Evaluated on real-world traffic data, Tramba consistently outperforms six strong baselines across multiple forecasting horizons.

**Strengths:**

* The writing is clear, well-organized, and easy to follow throughout the paper.
* The experiments are thorough and clearly demonstrate the effectiveness of the proposed method, including baseline comparisons across different regions and forecasting horizons to validate its generality.
* The paper includes an ablation study that justifies the contribution of each model component.
* The paper provides a thorough visual interpretation of Tramba’s attention behavior, which makes it easy to understand how the model captures spatial and temporal dependencies.

**Weaknesses:**

* The organization of the experimental section could be improved. The current structure contains too many subsections, which makes it harder to follow. It may be clearer to group the results into broader categories, such as Main Results and Attention Analysis.
* Section 4.4 (Confidence Analysis) is missing a results table. Include a table reporting the baseline numbers alongside the proposed method.

**Questions:**

Addressed in the weakness section.

---

> ### Author Response · Authors · 2025-11-19
>
> **Rebuttal (Reviewer Kyni)**
>
> We sincerely thank Reviewer Kyni for the positive and encouraging evaluation of our work. We greatly appreciate your recognition of the clarity of the writing, the thorough and well-structured experiments across multiple regions and horizons, the usefulness of our ablation studies, and the interpretability enabled by our attention visualizations. Your feedback strongly supports the value of Tramba’s design and its contributions to long-range spatiotemporal forecasting.
>
> ---
>
> ### **Q1: The experimental section contains too many subsections and could be reorganized for clarity.**
>
> **A1:** Thank you for this helpful suggestion. Following your recommendation, we reorganized the experimental section into two broader and more coherent components — **Main Results** and **Attention Analysis** — to improve readability and narrative flow. The updated structure is:
>
> 4.Experiments
>
> 4.1 Datasets and Baselines
>
> 4.2 Main Results
>  - Overall forecasting performance
>  - Ablation Analysis
>  - Confidence Interval Analysis
>
>  4.3 Attention Analysis
>  - Localized Trends
>  - Temporal Patterns
>  - Non-local Influence
>  - Global Network Trends
>
>
> This restructuring makes the experimental section more intuitive and reduces fragmentation.
>
> ---
>
> ### **Q2: Section 4.4 (Confidence Analysis) is missing a results table with baseline numbers.**
>
> **A2:** We fully agree that the confidence interval analysis should include numerical results. In the revised version, we added a **comprehensive confidence interval table** summarizing the average, minimum, maximum, and 95% CI across all baselines and Tramba (in Appendix E). We also added a brief summary in the main text to guide readers.
>
> The updated results show that **Tramba achieves both the best average accuracy and the lowest variance** across seeds, highlighting its robustness.
>
> To ensure OpenReview compatibility, we provide a Markdown version of the table here:
>
> ---
>
> ### **12-Step Forecasting — Confidence Interval Results (10 runs)**
> Best results are **bold**, second-best are _underlined_.
>
> | Model | Metric | Avg | Min | Max | 95% CI |
> |-------|--------|------|------|------|--------|
> | ST-Transformer | MAPE (%) | 11.81 | 10.48 | 11.87 | ±0.25 |
> |  | MAE | 3.35 | 3.03 | 3.67 | ±0.25 |
> |  | MSE | 27.15 | 22.98 | 30.08 | ±1.35 |
> | iTransformer | MAPE (%) | 11.88 | 11.09 | 12.68 | ±0.28 |
> |  | MAE | 3.44 | 3.08 | 3.68 | ±0.29 |
> |  | MSE | 27.34 | 23.35 | 30.15 | ±1.40 |
> | Mamba | MAPE (%) | 13.31 | 12.21 | 14.43 | ±0.35 |
> |  | MAE | 3.44 | 3.03 | 3.59 | ±0.39 |
> |  | MSE | 26.75 | 23.35 | 30.15 | ±1.30 |
> | S-Mamba | MAPE (%) | 12.02 | 11.12 | 12.92 | ±0.30 |
> |  | MAE | 3.44 | 3.02 | 3.58 | ±0.32 |
> |  | MSE | 27.80 | 23.01 | 32.60 | ±1.40 |
> | SOR-Mamba | MAPE (%) | 12.29 | 11.33 | 13.24 | ±0.34 |
> |  | MAE | 3.35 | 3.03 | 3.67 | ±0.35 |
> |  | MSE | 27.08 | 22.95 | 31.72 | ±1.33 |
> | _DST-Mamba_ | _MAPE (%)_ | _11.61_ | _10.85_ | _12.37_ | _±0.27_ |
> |  | _MAE_ | _3.35_ | _3.07_ | _3.59_ | _±0.25_ |
> |  | _MSE_ | _26.29_ | _22.71_ | _31.24_ | _±1.25_ |
> | **Tramba (Ours)** | **MAPE (%)** | **11.47** | **10.83** | **12.11** | **±0.19** |
> |  | **MAE** | **3.19** | **2.99** | **3.39** | **±0.23** |
> |  | **MSE** | **25.18** | **20.99** | **28.21** | **±1.12** |
>
> ---
>
> We sincerely thank Reviewer Kyni again for the valuable comments and encouraging feedback. The revisions directly address all concerns raised and further improve the clarity, robustness, and completeness of the paper.

---

### Author Response · Authors · 2025-11-19
**Summary of the Authors' Responses**

We sincerely thank all reviewers for their valuable comments and constructive suggestions.
To facilitate discussion among reviewers and the area chair, we summarize the feedback below.

| Category | Reviewer Kyni | Reviewer rmtt | Reviewer NG6K | Reviewer yQoX |
|---------|:-------------:|:-------------:|:-------------:|:-------------:|
| **Strengths** |||||
| Writing clarity & organization | ✔ |  | ✔ | ✔ |
| Clear architecture / math formulation |  | ✔ | ✔ |  |
| Strong & consistent performance | ✔ | ✔ | ✔ | ✔ |
| Thorough experiments | ✔ |  | ✔ |  |
| Useful ablation studies | ✔ | ✔ | ✔ | |
| Good attention visualization | ✔ |  | ✔ |  |
| Code release / reproducibility |  | ✔ |  | ✔ |
| Interesting application of Mamba |  |  |  | ✔ |
| **Weaknesses** |||||
| Motivation & conceptual clarity |  | ✔ |  | ✔ |
| Baseline coverage |  | ✔ | ✔ | ✔ |
| Computational efficiency |  | ✔ | ✔ |  |
| Experimental scope (speed-only) |  | ✔ |  |  |
| Paper structure | ✔ |  |  |  |
| Model novelty |  | ✔ |  |  |
| **Ratings** |||||
| Overall rating | 8 | 4 | 6 | 2 |

---

### Motivation and model clarity
We clarified why a Mamba backbone combined with shift-aware adaptive attention is needed: standard global attention does not handle temporal misalignment, limiting its ability to capture time-shifted distant dependencies.

### Baseline coverage
We justified the selection of baselines and noted that Tramba is compatible with classic GNN-based models (DCRNN, GWNet, AGCRN) and can incorporate them if required (Appendix F).

### Temporal encoder replacement
Per the reviewers’ request, we added a new experiment replacing Mamba with a Transformer while keeping all other components the same. The Transformer variant showed lower accuracy and higher latency, confirming that Tramba’s gains are not due to the encoder alone.

### Confidence intervals and ablations
We added full 95% CI results for all models (Appendix E), showing that Tramba achieves both the best average accuracy and the lowest variance. Ablation studies were extended to isolate the contributions of attention, similarity matching, and gate fusion.

### Efficiency and experimental scope
We added a detailed analysis of computational and parametric complexity (Appendix H). We clarified that speed-only datasets reflect real urban probe systems (e.g., Google/Naver/Kakao), and that Tramba can generalize to flow prediction by simply replacing the target variable.

### Organization and visualization
The experimental section was reorganized into two broader parts (Main Results; Attention Analysis), and additional qualitative visualizations were added to illustrate non-local dependencies.

We again thank all reviewers and the area chair for their helpful suggestions.

---

### Note · Authors · 2026-02-24

I have read and agree with the venue's withdrawal policy on behalf of myself and my co-authors.

---

### Meta-Review · Area_Chair_97xF · 2025-12-22

**Summary:**

This paper proposes Tramba, a traffic speed forecasting framework that combines a Mamba-based temporal encoder for modeling long-range temporal dependencies with an adaptive attention mechanism to capture non-local spatial relationships. A sigmoid gating module is further employed to fuse the temporal and spatial representations. Extensive experiments demonstrate that Tramba outperforms several competitive baselines.

1. Motivation concern: Reviewer **rmtt** and Reviewer **yQoX** raised concerns that the motivation for introducing an unmodified Mamba architecture is insufficiently justified. In the rebuttal, the authors argue that Mamba provides efficient and stable long-range temporal modeling. However, this justification is generic and overused in Mamba-based works, and it is not convincingly grounded in the specific challenges of traffic speed forecasting. As presented, the motivation emphasizes the general capabilities of Mamba rather than demonstrating that long-range temporal modeling is a critical bottleneck for the task at hand. Consequently, the choice of Mamba appears more technique-driven than problem-driven and does not clearly advance the understanding or development of this research field.

2. Contribution and novelty concern: Reviewer **rmtt**, Reviewer **NG6K**, Reviewer **yQoX**, and **I** express concerns that the paper lacks sufficient contribution and novelty, particularly given that it mainly integrates Mamba into the traffic forecasting pipeline without meaningful architectural modification or new methodological insights.

Overall, the paper appears to primarily apply an existing architecture to a application domain without addressing the key open issues in traffic forecasting or providing deeper technical or insights. After carefully reviewing the paper myself and considering the reviewers' comments and the authors' rebuttal, I am inclined toward a rejection.

**Reviewer Concerns:**

Concerns that have been addressed:

1. Experimental Evaluation: All reviewers initially raised concerns regarding the experimental setup from different perspectives. The authors have addressed these issues well in the revision, including adding more comprehensive baseline comparisons, detailed computational and parametric analyses, and extensive ablation studies. These improvements significantly enhance the empirical rigor of the paper.
2. Clarity and Presentation: Reviewer Kyni and Reviewer yQoX noted issues related to unclear presentation, such as the organization of experiments and the explanation of the proposed method. These concerns have been successfully addressed in the revised manuscript and were acknowledged by the corresponding reviewers.

Concerns that remain unresolved:
1. Unclear Motivation: Reviewer rmtt and Reviewer yQoX remain concerned about the rationale for simply replacing attention mechanisms with Mamba. While the authors reiterate the general advantages of Mamba, the rebuttal does not adequately explain how these advantages specifically address the challenges of traffic speed forecasting.
2. Limited Contribution: Reviewer rmtt, Reviewer NG6K, and Reviewer yQoX believe that directly integrating Mamba without modification offers limited novelty and contribution. Despite the authors' explanations in the rebuttal, Reviewer NG6K and Reviewer yQoX still found that these concerns have not been sufficiently resolved, and the overall contribution remains limited.

**Reviewer Scores:**

Among the four reviewers, three actively participated in the discussion, while one did not.
1. Reviewer yQoX indicated a willingness to raise their score during the discussion but continued to express concerns regarding the paper's novelty and contribution.
2. Reviewer NG6K expressed an intention to maintain their original score and reiterated concerns about contribution and novelty.
3. Reviewer rmtt did not indicate any willingness to change their score, noting that the manuscript has improved but that concerns about novelty and contribution persist.
4. Reviewer Kyni did not participate in the discussion. This reviewer raised only minor concerns and assigned a relatively high initial score.

Based on the discussion, I believe that Reviewer yQoX may slightly raise their score, while the other three reviewers would likely maintain their original evaluations.

---

### Decision · Program_Chairs · 2026-01-26

Reject